# Graph-based Deterministic Policy Gradient for Repetitive Combinatorial Optimization Problems

**Zhongyuan Zhao**
Rice University
zhongyuan.zhao@
rice.edu

**Ananthram Swami**
DEVCOM Army Research Laboratory
ananthram.swami.civ@army.mil

**Santiago Segarra**
Rice University
segarra@rice.edu

## Abstract

We propose an actor-critic framework for graph-based machine learning pipelines with non-differentiable blocks, and apply it to repetitive combinatorial optimization problems (COPs) under hard constraints. Repetitive COP refers to problems to be solved repeatedly on graphs of the same or slowly changing topology but rapidly changing node or edge weights. Compared to one-shot COPs, repetitive COPs often rely on fast (distributed) heuristics to solve one instance of the problem before the next one arrives, at the cost of a relatively large optimality gap. Through numerical experiments on several discrete optimization problems, we show that our approach can learn reusable policies to reduce the optimality gap of fast (distributed) heuristics for independent repetitive COPs, and can optimize the long-term objectives for repetitive COPs embedded in graph-based Markov decision processes. Source code at `https://github.com/XzrTGMu/twin-nphard`.

## 1 Introduction

In a general network setting, the network state is captured by a graph $\mathcal{G} = (\mathcal{V}, \mathcal{E}, \mathbf{S})$, that comprises a vertex set $\mathcal{V}$, an edge set $\mathcal{E}$, and a matrix $\mathbf{S}$ capturing the node features. A vector $\mathbf{o} \in \mathbb{R}^{|\mathcal{V}|}$ captures the outcomes on individual nodes of a non-differentiable network process, $f_{net}(\cdot)$, as:

$$\mathbf{o} = f_{net}(\mathcal{G}) . \tag{1}$$

We aim to optimize the system-level objective $f_{obj}(\mathbf{o})$, where $f_{obj} : \mathbb{R}^{|\mathbf{o}|} \to \mathbb{R}$ is a known linear combination, by improving parts of the network process $f_{net}(\cdot)$. In particular, we are interested in network processes that involve graph-based repetitive combinatorial optimization problems (R-COPs) (Kraay & Harker, 1996) – COPs that need to be solved repeatedly on graphs of the same or slowly changing topology ($\mathcal{V}$ or $\mathcal{E}$) but rapidly changing node features ($\mathbf{S}$), under hard constraints that must be satisfied at all times (Kendall, 1975), which often make COPs NP-hard. R-COPs have many real-world applications, such as task scheduling (Pinedo, 2012), route planning (Vogiatzis & Pardalos, 2013), link scheduling (Joo & Shroff, 2012; Paschalidis et al., 2015; Eisen et al., 2019; Zhao et al., 2022a;b) and routing (Oliveira et al., 2011) in communication networks, and energy management in smart grids (Chau et al., 2018), where a node or edge weight captures varying cost or utility.

In practice, solvers for R-COPs are often subject to restricted runtime and/or distributed execution. For example, in scheduling, network routing, and multi-object tracking in computer vision, COPs need to be solved within tens of milliseconds to a few seconds. Although memory-based approaches can avoid solving each instance from scratch for some applications (Kraay & Harker, 1996; Wang, 2021), in general, R-COPs rely on fast heuristics to meet the strict time constraints, at the cost of relatively large optimality gaps. Moreover, in real-time networked systems, such as communication networks, smart grids, and robot swarms (Tolstaya et al., 2020), centralized solvers often suffer from the large communication overhead of gathering the full network state, high computational complexity, and risk of single point of failure. Therefore, distributed solutions (Moser & Tardos, 2010) are preferred for better scalability and robustness, in which nodes across the network work in parallel to collect and process the information of only their local neighborhoods for decision making.

Depending on the definition of $f_{obj}$, R-COPs can be categorized as, 1) independent R-COPs, i.e., $\mathbf{S}(t_1)$ and $\mathbf{S}(t_2)$ are considered as independent if $t_1 \neq t_2$, and 2) R-COPs embedded in a graph-based

Markov decision process (MDP). For example, in link scheduling for wireless multiple networks (Zhao et al., 2022a;b), the network state $\mathcal{G}(t) = (\mathcal{V}, \mathcal{E}, \mathbf{S}(t))$, where $\mathbf{S}(t)$ captures the packet backlogs of all the links at time $t$, depends on the schedule of $t-1$ found by solving a maximum weight independent set (MWIS) problem defined on $\mathcal{G}(t-1)$. Scheduling for maximum throughput (number of data packets transmitted on the schedule) is equivalent to optimize each MWIS instance individually (Zhao et al., 2022a), which is formulated as an independent R-COP. However, to minimize the average backlog (packets left in the queues by the schedule) across links and *over time*, the transition of network states must be considered and the scheduling is formulated as R-COP in a graph-based MDP (Zhao et al., 2022b). Similar MDP formulations can also be found in wireless scheduling for battery lifetime (Sikandar et al., 2020), vehicle routing for waste collection (Wu et al., 2020), and inventory control in distribution networks (Çelebi, 2015). However, the 2nd type of R-COPs have rarely been addressed, except in an ad-hoc manner for link scheduling (Zhao et al., 2022b). Therefore, a general approach to R-COPs in graph-based MDPs would be of high interest.

## 1.1 Existing Approaches to COPs

**Centralized solvers:** The general approach for exactly solving a COP is to formulate it as a mixed-integer program, and solve it by branch-and-bound (Land & Doig, 1960) or dynamic programming. Commercial Gurobi solvers (LLC, 2020) can exactly solve COPs on graphs of hundreds of nodes in reasonable time. Although large real-world graphs with certain structural properties can be reduced (Lamm et al., 2019) to find exact solutions, problems of large scale and/or stringent time limits often rely on efficient heuristics to approximate the solutions; common strategies include greedy algorithms, local search (Wang et al., 2018), tabu search, ant colony (Jovanovic et al., 2010), and simulated annealing. In recent machine learning-based COP solvers, graph neural networks (GNNs) (Wu et al., 2021) are trained to guide an algorithmic framework, which guarantees the constraints being always followed; common frameworks include branching (Khalil et al., 2016; Gasse et al., 2019; Nair et al., 2020; Zarpellon et al., 2021), tree search (Li et al., 2018), greedy algorithms (Khalil et al., 2017; Zhao et al., 2022a), and local search (Hudson et al., 2022). In these approaches, a COP instance is formulated as a finite episode of an MDP, with the residual graph of each intermediate step defined as a state, based on which an action of adding one vertex in the residual graph to the partial solution is generated by a GNN, and a solution is built from a sequence of such scalar actions. This formulation has a time complexity of at least $\mathcal{O}(|\mathcal{V}||\mathcal{E}|)$, since a GNN of time complexity of $\mathcal{O}(|\mathcal{E}|)$ (Wu et al., 2021) is called in each intermediate step. (Drori et al., 2020; Hottung et al., 2022) lowered the time complexity of vehicle routing problems to $\mathcal{O}(|\mathcal{E}|)$ (typically, $|\mathcal{E}| > |\mathcal{V}|$) by encoding the graph by a GNN only once per COP instance. However, for R-COPs, this formulation is limited by centralized execution and the complexity of a GNN, and does not apply to R-COPs in graph-based MDPs.

**Distributed solvers:** Popular distributed approaches to COPs include Moser-Tardos algorithm (Moser & Tardos, 2010) and dynamic programming. The complexity of distributed algorithms is typically measured by local communication complexity, which refers to the rounds of message exchanges between a node and its neighbors (Joo & Shroff, 2012; Zhao et al., 2022a). For example, on MWIS, Moser-Tardos algorithm (Joo & Shroff, 2012; Moser & Tardos, 2010) can converge in $\mathcal{O}(\log^* |V|)$ rounds, whereas distributed dynamic programming (Paschalidis et al., 2015) in $\mathcal{O}(|V|)$ rounds. Distributed algorithms may have larger optimality gaps due to the lack of global information, but topological metrics, such as edge betweenness for the Steiner tree problem(Fujita et al., 2016), can help reduce the gap. Learning-based distributed COP solvers have received less attention compared to their centralized counterparts. GCN-LGS (Zhao et al., 2022a) is an architecture proposed for a repetitive MWIS problem, in which an $L$-layer graph convolutional neural network (GCNN) generates a vector $\mathbf{z}$ for a topology $(\mathcal{V}, \mathcal{E})$, and a greedy heuristic solves modified instances $(\mathcal{V}, \mathcal{E}, \mathbf{c}(t) \odot \mathbf{z})$, rather than the original ones $(\mathcal{V}, \mathcal{E}, \mathbf{c}(t))$, for the next $N$ time slots $t \in \{1, \ldots, N\}$, assuming the topology would not change by $t = N$. GCN-LGS can reduce the optimality gap of greedy solvers (Joo & Shroff, 2012) by ⅓ to ½, with a time complexity of $\mathcal{O}(L|\mathcal{E}|/N + |\mathcal{V}|)$, or can converge in $\mathcal{O}(L/N + \log |V|)$ iterations in a distributed setting. For large reusing factor $N$, the overhead of GCNN is negligible. However, the training methods in (Zhao et al., 2022a;b) are ad hoc and specific to MWIS problems, and may not be directly applicable to other types of R-COPs.

In this work, we generalize the GCN-LGS architecture in (Zhao et al., 2022a;b) to a wider range of R-COPs, through a unified framework of problem formulation and actor-critic architecture. Unlike previous works that solve a COP instance through a sequence of scalar actions, our actor network generates a high-dimensional *intermediate action* to parameterize a given classical heuristic, $h'(\cdot)$, which produces vectorized decisions by solving one or multiple instances of an R-COP. This formula-

tion allows us to pick a fast and/or distributed classical heuristic $h'(\cdot)$ that adheres to various practical restrictions, while guaranteeing the decisions always follow the hard constraints. For independent R-COPs, an intermediate action encodes the underlying topology shared by $N$ instances to improve the average quality of their solutions, reducing the GNN overhead from $\mathcal{O}(|\mathcal{E}|)$ (Drori et al., 2020; Hottung et al., 2022) to $\mathcal{O}(|\mathcal{E}|/N)$, which is arbitrarily small for a large $N$. In R-COPs embedded in a graph-based MDP, an intermediate action for the optimal expected long-term objective is generated based on the network state in each time step, serving as the cost vector of the corresponding COP instance, which is later translated into a decision vector by $h'(\cdot)$. The challenge, however, is that our policy network contains a non-differentiable block, e.g., $h'(\cdot)$.

## 1.2 Learning in Non-differentiable Pipelines

Training with supervised or unsupervised learning methods is challenging in the presence of non-differentiable blocks. Reinforcement learning (RL) (Sutton & Barto, 2018) addresses this problem by treating the non-differentiable block as (part of) the environment (Silver et al., 2014; Khalil et al., 2017; Zhao et al., 2022a). Compared to Q-learning (Watkins & Dayan, 1992) that is sequential, policy gradient (Silver et al., 2014) is better suited for the high-dimensional action spaces in networks of parallel and dynamic nature. A major issue in RL for networks is how to assign credit to individual elements based on the system-level reward. Zeroth-order optimization (ZOO) (Liu et al., 2020) is the last resort for non-differentiable pipelines, as it requires numerous evaluations of $f_{net}(\cdot)$, which can be computationally prohibitive. In some scenarios, soft constraints (Kendall, 1975) can avoid the use of non-differentiable pipeline, allowing a fully differentiable policy network being trained by primal-dual optimization (Eisen et al., 2019) or imitation learning (Ross et al., 2011).

To address the aforementioned shortcomings, we propose GDPG-Twin, a graph-based deterministic policy gradient method based on the actor-critic framework (Sutton & Barto, 2018, ch. 13). GDPG-Twin trains a differentiable twin of the non-differentiable policy block in $f_{net}(\cdot)$, e.g., $h'(\cdot)$, as part of the critic network to facilitate the training of the actor network. A similar differentiable approximation bridges (DAB) (Ramapuram & Webb, 2020) can approximate the *immediate* behavior of a non-differentiable block in standalone systems. We improve DAB from three aspects: we use a twin network to predict the element-wise *expected* outcomes of the non-differentiable policy network, use a GNN to account for the permutation equivariance in network settings, and use random policy sampling for static policy parameters $\mathbf{Z}$ in a fixed topology. In addition, GDPG-Twin is more efficient than ZOO. Although we focus on node-related problems, GDPG-Twin also applies to edge-related problems, e.g., by using simplicial neural networks (Roddenberry et al., 2021).

## 1.3 Contributions

In summary, our proposed framework: 1) can generalize to learning for different COPs without handcrafting the credit assignment strategies as in other schemes of network-based RL (Eisen et al., 2019; Zhao et al., 2022a;b), 2) works for R-COPs with hard constraints, 3) requires fewer evaluations of $f_{net}(\cdot)$ than ZOO (Liu et al., 2020), and 4) has the advantage of RL schemes in not relying on expensive data labeling as in supervised learning or a computationally intensive supervising algorithm as in imitation learning (Ross et al., 2011). Our contribution contains the following three aspects:

- We propose a general approach to R-COPs under hard constraints and practical restrictions. Our approach can reduce the optimality gap of fast and/or distributed heuristics for independent R-COPs, at the cost of only an upfront computation and communication overhead. Moreover, it can optimize the long-term objectives for R-COPs in a graph-based MDP by embedding the future reward into the cost vector of COP instance at each time step.

- We propose GDPG-Twin, an actor-critic architecture for network settings. By using a twin network that learns the element-wise expected outcomes $\mathbf{o}$ of a non-differentiable policy network in $f_{net}(\cdot)$, the critic can leverage the knowledge of the linear combination of the system-level objective $f_{obj}(\mathbf{o})$ to address the challenge of credit assignment across the network, which is a major roadblock for discrete or mixed-integer network processes.

- We adopt a random policy sampling strategy in the training of the twin network, which enables optimizing a static policy for R-COPs defined on fixed topologies. Moreover, our approach requires significantly fewer evaluations of the network process $f_{net}(\cdot)$ than ZOO.

**Notation.** Upright bold lower(upper)-case symbols are used to denote column vectors, e.g., $\mathbf{x}$ (matrices, e.g., $\mathbf{X}$). $\mathbf{x}_i$ denotes the $i$th element of vector $\mathbf{x}$, $\mathbf{X}_{i,j}$ denotes the element at row $i$ and

column $j$ of matrix $\mathbf{X}$, $\mathbf{X}_{i*}$ (or $\mathbf{X}_{*j}$) denotes row $i$ (or column $j$) of matrix $\mathbf{X}$. Unless otherwise specified, calligraphic upper-case symbols (e.g., $\mathcal{V}$) are used to denote sets. $(\cdot)^\top$ and $\odot$ denote transpose and element-wise product, respectively.

## 2 PROBLEM FORMULATION

Many binary discrete COPs have the following formulation,

$$\mathbf{x}^* = \min_{\mathbf{x}} \ \mathbf{c}^\top \mathbf{x} \tag{2a}$$

$$s.t. \ \ \mathbf{x}_i \in \{0, 1\} \,, \forall i \in \{1, \dots, |\mathcal{V}|\} \,, \tag{2b}$$

$$\text{other problem-specific constraints,} \tag{2c}$$

where $\mathbf{x}$ and $\mathbf{c}$ are respectively the vectors of decisions and weights on nodes, and constraints in (2c) are often defined on the graph, e.g., $\mathbf{x}_i + \mathbf{x}_j \leq 1, \forall \{i, j\} \in \mathcal{E}$. Without loss of generality, (2b) can be of other arities and (2c) can be on a hypergraph or simplicial complex. Since many problems in (2) are NP-hard, we seek to develop efficient heuristics to approximate $\mathbf{x}^*$. Furthermore, we define the function space $\mathcal{F}$ of *valid* heuristics, meaning that $\mathbf{x} = f(\mathcal{V}, \mathcal{E}, \mathbf{c})$ satisfies the constraints in (2) for all $f \in \mathcal{F}$. We also define the space $\mathcal{P}$ of *practical* functions, i.e., functions that satisfy pre-specified practical restrictions, e.g., limited runtime and/or distributed execution.

### 2.1 INDEPENDENT REPETITIVE COPS

For independent R-COPs, the weight $\mathbf{c}$ of an instance is considered to be a random vector drawn from its target sampling distribution $\Omega^{\mathbf{c}}$. We formulate independent R-COPs as finding a valid heuristic (policy) that optimizes the expectation of the objective function in (2a) under practical restrictions

$$h^* = \min_{h \in (\mathcal{F} \cap \mathcal{P})} \mathbb{E}_{\mathbf{c} \sim \Omega^c}(\mathbf{c}^\top \mathbf{x}) \tag{3a}$$

$$s.t. \ \ \mathbf{x} = h(\mathcal{V}, \mathcal{E}, \mathbf{c}) \,. \tag{3b}$$

### 2.2 REPETITIVE COPS IN A GRAPH-BASED MARKOV DECISION PROCESS

Given network state $\mathcal{G}(t) = (\mathcal{V}(t), \mathcal{E}(t), \mathbf{S}(t))$, where $\mathbf{S}(t)$ captures features on nodes at time $t$, our objective is to find a valid heuristic $h$ for a COP in (2) and a cost function $\Psi$, which together form a policy that – subject to the practical restrictions – maps $\mathcal{G}(t)$ into $\mathbf{x}(t)$ and maximizes the expected system-level value over time horizon $T$, as follows:

$$h^*, \Psi^* = \max_{h \in (\mathcal{F} \cap \mathcal{P}), \Psi \in \mathcal{P}} \mathbb{E}_{\mathcal{G}(1) \sim \Omega^1} \left[ f_{obj} \left( \mathbf{o}(1) \right) \right] \,, \tag{4a}$$

$$s.t. \ \ \mathbf{o}(t) = \mathbb{E}_{\{h, \Psi\}} \left[ \sum_{k=0}^{T-t} \gamma^k \mathbf{r}(t+k) \middle| \mathcal{G}(t) \right] \,, \tag{4b}$$

$$\mathbf{r}(t) = f_r(\mathcal{V}(t), \mathcal{E}(t), \mathbf{S}(t), \mathbf{x}(t)) \,, \tag{4c}$$

$$\mathbf{x}(t) = h(\mathcal{V}(t), \mathcal{E}(t), \mathbf{c}(t)) \,, \tag{4d}$$

$$\mathbf{c}(t) = \Psi(\mathcal{V}(t), \mathcal{E}(t), \mathbf{S}(t)) \,, \tag{4e}$$

$$\mathbf{S}(t+1) = f_s(\mathcal{V}(t), \mathcal{E}(t), \mathbf{S}(t), \mathbf{x}(t)) \,. \tag{4f}$$

In (4), $\mathbf{o}(t)$ is the value vector of current state $\mathcal{G}(t)$ under policy $\{h, \Psi\}$, $0 \leq \gamma \leq 1$ is the discount factor, (4a) states that the system objective is the expectation of a known linear combination, $f_{obj} : \mathbb{R}^{|\mathbf{o}|} \to \mathbb{R}$, of $\mathbf{o}(1)$ over the initial state distribution $\Omega^1$, capturing both average reward and start-state formulations (Sutton et al., 1999), (4d) states that a decision vector $\mathbf{x}(t)$ is generated by a valid heuristic $h \in \mathcal{F}$ of a COP based on the network topology $(\mathcal{V}(t), \mathcal{E}(t))$ and cost vector $\mathbf{c}(t)$, which is a function of the network state $\mathcal{G}(t)$ as stated by (4e), and (4c) and (4f) define the MDP by respectively stating that the reward vector $\mathbf{r}(t)$ and the next state $\mathbf{S}(t+1)$ depend on the current state $\mathbf{S}(t)$ and the decisions $\mathbf{x}(t)$. In general, $f_r(\cdot)$ and $f_s(\cdot)$ in (4) are stochastic functions, capturing some stationary random processes in the environment. The formulation in (4) can capture R-COPs with accumulative objectives, such as latency and battery lifetime in wireless scheduling, waste level in vehicle routing for waste collection, and inventory in distribution networks. Notice that if $T = t$, $\gamma = 1$, $\mathbf{S}(t) = \mathbf{c}(t)$ ($\Psi$ is bypassed), $\mathbf{r}(t) = \mathbf{c}(t) \odot \mathbf{x}(t)$, $f_{obj}(\mathbf{o}) = \mathbf{1}^\top \mathbf{o}$, and we set $f_s$ in (4f) as drawing a random vector from $\Omega^c$, then (4) boils down to (3). Thus, (4) is a generalized form of all R-COPs. Appendix A further illustrate (3) and (4) via exemplary formulations of two wireless scheduling problems.

## 3 GRAPH-BASED DETERMINISTIC POLICY GRADIENT

Our downstream pipeline follows the existing methodology of using a neural network to guide a discrete operation, which guarantees the hard constraints, but keeps the neural network outside the iterations of the algorithmic framework for lower complexity (Zhao et al., 2022a). In this section, we provide the main solution, whereas the full procedures of optimization for Sections 3.1 and 3.2 are given by Algorithms 1 and 2 in Appendix D. Since (3) and (4) are functional optimization problems, we seek to approximately solve them by parameterizing the policy ($h$ or $\{h, \Psi\}$) and reformulating (3) and (4) as finding the optimal set of parameters. To meet the practical restrictions, we rely on manual selections (or design) of a baseline heuristic $h' \in \mathcal{F}$ and the parameterizations.

### 3.1 LEARNING FOR INDEPENDENT REPETITIVE COP

For independent R-COPs, we reformulate (3) as

$$\mathbf{Z}^* = \min_{\mathbf{Z} \in \mathbb{R}^{|\mathbf{c}| \times g}} \mathbf{1}^\top \mathbf{o} \tag{5a}$$

$$s.t. \quad \mathbf{o} = \mathbb{E}_{\mathbf{c} \sim \Omega^{\mathbf{c}}} (\mathbf{c} \odot \mathbf{x}) \tag{5b}$$

$$\mathbf{x} = h'(\mathcal{V}, \mathcal{E}, \mathbf{w}) , \tag{5c}$$

$$\mathbf{w}_i = f_{loc}(\mathbf{c}_i; \mathbf{Z}_{i*}) , \forall i \in \{1, \ldots, |\mathbf{c}|\} . \tag{5d}$$

The objectives in (3a) and (5a) are equal due to the linearity of expectation, i.e., $\mathbb{E}_{\mathbf{c} \sim \Omega^{\mathbf{c}}}(\mathbf{c}^\top \mathbf{x}) = \mathbb{E}_{\mathbf{c} \sim \Omega^{\mathbf{c}}}(\sum_{i=1}^{|\mathbf{c}|} \mathbf{c}_i \mathbf{x}_i) = \sum_{i=1}^{|\mathbf{c}|} \mathbb{E}_{\mathbf{c} \sim \Omega^{\mathbf{c}}}(\mathbf{c}_i \mathbf{x}_i) = \mathbf{1}^\top \mathbf{o}$. Constraints (5c) and (5d) further break the parameterized policy in (3b), $h(\cdot; \mathbf{Z})$, into a baseline heuristic $h'(\cdot) \in (\mathcal{F} \cap \mathcal{P})$ given in advance and a parameterized local function $f_{loc}(\cdot; \mathbf{Z}_{i*})$, where $\mathbf{Z}_{i*} \in \mathbb{R}^{1 \times g}$ captures the $g$ local parameters for node $i \in \mathcal{V}$. The local function $f_{loc}(\cdot; \mathbf{Z}_{i*})$ can be chosen as, e.g., a multiplier, a single neuron, or even a small neural network, and depends on node-specific parameters $\mathbf{Z}_{i*}$.

To solve the problem formulated in (5), we employ deterministic policy gradient reinforcement learning, where the policy parameters are $\mathbf{Z}$. However, the gradient $\nabla_{\mathbf{Z}} \mathbf{1}^\top \mathbf{o}$ is not available since $h'(\cdot)$ is non-differentiable. To address this problem, we introduce a trainable and differentiable twin network $f_{twin}(\cdot; \mathbf{\Theta}_c)$ to learn the element-wise expected outcome of $h(\cdot; \mathbf{Z})$. In contrast to the critic in a typical standalone setting, which directly predicts the system-level objective, the twin network works for network settings, where the policy parameters and outcome are supported on graphs. The twin can be implemented by a graph or a simplicial neural network, depending on the specific COP. The expected behavior of the twin can be described by

$$\mathbf{o} \approx \hat{\mathbf{o}} = f_{twin}(\mathcal{V}, \mathcal{E}, \bar{\mathbf{c}}, \mathbf{Z}; \mathbf{\Theta}_c^*), \text{ where } \bar{\mathbf{c}} = \mathbb{E}_{\Omega^{\mathbf{c}}}(\mathbf{c}) . \tag{6}$$

Based on (5a) and (6), we can estimate the system objective as $\mathbf{1}^\top \hat{\mathbf{o}}$. Based on (6) and the chain rule $\frac{\partial \mathbf{1}^\top \hat{\mathbf{o}}}{\partial \mathbf{Z}} = \mathbf{1}^\top \frac{\partial \hat{\mathbf{o}}}{\partial \mathbf{Z}}$, the policy gradient is estimated as (Silver et al., 2014)

$$\widehat{\nabla_{\mathbf{Z}} \mathbf{1}^\top \mathbf{o}} \approx \nabla_{\mathbf{Z}} \mathbf{1}^\top \hat{\mathbf{o}} = \nabla_{\mathbf{Z}} f_{twin}(\mathcal{V}, \mathcal{E}, \mathbb{E}_{\Omega^{\mathbf{c}}}(\mathbf{c}), \mathbf{Z}; \mathbf{\Theta}_c) \mathbf{1} . \tag{7}$$

Given a policy learning rate $0 \leq \alpha_p \leq 1$, we can update the policy parameters as, $\mathbf{Z} \leftarrow \mathbf{Z} - \alpha_p \nabla_{\mathbf{Z}} \mathbf{1}^\top \hat{\mathbf{o}}$. For applications on static topologies, we can optimize $\mathbf{Z}$ directly with stochastic gradient descent.

For R-COPs on dynamic networks, we want the policy parameters to be a function of the topology, implemented as an actor, $\mathbf{Z} = \Psi(\mathcal{V}, \mathcal{E}, \bar{\mathbf{c}}; \mathbf{\Theta}_p)$. In this case, we can estimate the gradient $\nabla_{\mathbf{\Theta}_p} \mathbf{1}^\top \hat{\mathbf{o}} = \nabla_{\mathbf{\Theta}_p} \Psi(\mathcal{V}, \mathcal{E}, \bar{\mathbf{c}}; \mathbf{\Theta}_p) \nabla_{\mathbf{Z}} \mathbf{1}^\top \hat{\mathbf{o}}$, and update the actor parameters as, $\mathbf{\Theta}_p \leftarrow \mathbf{\Theta}_p - \alpha_p \nabla_{\mathbf{\Theta}_p} \mathbf{1}^\top \hat{\mathbf{o}}$. From the perspective of the actor $\Psi(\cdot; \mathbf{\Theta}_p)$, its input is $(\mathcal{V}, \mathcal{E}, \bar{\mathbf{c}})$ instead of the instantaneous network state $(\mathcal{V}, \mathcal{E}, \mathbf{c})$, and its output is an intermediate action $\mathbf{Z}$, which is used as the policy parameters.

Given a learning rate $0 \leq \alpha_c \leq 1$, the twin network can be updated by the following gradient descent,

$$\mathbf{\Theta}_c \leftarrow \mathbf{\Theta}_c - \alpha_c \nabla_{\mathbf{\Theta}_c} \ell_{mse}(\hat{\mathbf{o}}, \mathbf{o}) , \tag{8}$$

where the loss function $\ell_{mse}(\hat{\mathbf{o}}, \mathbf{o})$ is the mean-square-error (MSE) between $\hat{\mathbf{o}}$ and $\mathbf{o}$. Since $\mathbf{o}$ is an expectation over sampling space $\Omega^{\mathbf{c}}$, we can implement the following stochastic gradient descent

$$\mathbf{\Theta}_c \leftarrow \mathbf{\Theta}_c - \alpha_c \nabla_{\mathbf{\Theta}_c} \ell_{mse}(\hat{\mathbf{o}}, \mathbf{c} \odot \mathbf{x}), \ \ell_{mse}(\hat{\mathbf{o}}, \mathbf{c} \odot \mathbf{x}) = \frac{1}{|\mathbf{x}|} \sum_{i=1}^{|\mathbf{x}|} (\hat{\mathbf{o}}_i - \mathbf{c}_i \mathbf{x}_i)^2 , \ \mathbf{c} \in \Omega^{|\mathbf{c}|} , \quad (9)$$

by minimizing the MSE loss in (9) with an off-the-shelf optimizer. Notice that we need to evaluate $h'(\cdot)$ and $f_{loc}(\cdot; \mathbf{Z})$ to get $\mathbf{x}$ in (9). A detailed derivation of (9) is given in Appendix B.

### 3.1.1 RANDOM SAMPLING AROUND CURRENT POLICY

For applications based on static topologies, i.e., $(\mathcal{V}, \mathcal{E}, \bar{\mathbf{c}})$ are constant, we no longer need an actor to generate $\mathbf{Z}$. In this case, the twin is likely to be overfitted if we only feed it with a static $\mathbf{Z}$ during training. To address this problem, we feed the twin $f_{twin}(\cdot)$ and $h(\cdot; \mathbf{Z})$ with random samples around the current policy parameters $\mathbf{Z}^{(j)} = \mathbf{Z} + \mathbf{N}^{(j)}, \mathbf{N}_{m,n}^{(j)} \in \mathbb{U}(-\epsilon, \epsilon)$ where $\epsilon$ is the sampling radius. The loss in (9) then becomes $\ell_{mse}(\hat{\mathbf{o}}^{(j)}, \mathbf{c} \odot \mathbf{x}^{(j)})$. This random sampling strategy enables the critic, comprising the twin $f_{twin}(\cdot; \mathbf{\Theta}_c)$ and system-level objective function $f_{obj}(\hat{\mathbf{o}})$, to learn the loss landscape around the current $\mathbf{Z}$, thus improve the quality of gradient in (7).

In ZOO, a policy gradient is estimated from at least two random samples around the current $\mathbf{Z}$ (including $\mathbf{Z}$ itself) (Liu et al., 2020). While fewer policy samples requires fewer evaluations of $h(\cdot; \mathbf{Z})$, which could be computationally expensive in many applications, it could degrade the convergence by raising the noise floor of the gradient estimate. Compared to ZOO, our twin-based critic can continuously learn, refine, and memorize the loss landscape around the current policy as new samples coming in, leading to better gradient estimate in backpropagation (improved convergence as shown in Figure. 1), and higher policy sampling efficiency, as shown in Sections 4.2 and 4.3.

### 3.2 LEARNING FOR REPETITIVE COP IN A GRAPH-BASED MARKOV DECISION PROCESS

For R-COP in a graph-based MDP, we parameterize the policy in (4), $\{h, \Psi\}$, with a given baseline heuristic $h' \in \mathcal{F}$, and a parameterized cost function $\Psi(\cdot; \mathbf{\Theta}_p)$. We then reformulate (4) as:

$$\mathbf{\Theta}_p^* = \max_{\mathbf{\Theta}_p \in \mathbb{R}^{|\mathbf{\Theta}_p|}} \mathbb{E}_{\mathcal{G}(1) \sim \Omega^1} \left[ f_{obj} \left( \mathbf{o}(1) \right) \right] , \tag{10a}$$

$$s.t. \ \mathbf{o}(t) = \mathbb{E}_{\mathbf{\Theta}_p} \left[ \sum_{k=0}^{T-t} \gamma^k \mathbf{r}(t+k) \Big| \mathcal{G}(t) \right] , \tag{10b}$$

$$\mathbf{r}(t) = f_r(\mathcal{V}(t), \mathcal{E}(t), \mathbf{S}(t), \mathbf{x}(t)) , \tag{10c}$$

$$\mathbf{x}(t) = h'(\mathcal{V}(t), \mathcal{E}(t), \mathbf{c}(t)) , \tag{10d}$$

$$\mathbf{c}(t) = \Psi(\mathcal{V}(t), \mathcal{E}(t), \mathbf{S}(t); \mathbf{\Theta}_p) , \tag{10e}$$

$$\mathbf{S}(t+1) = f_s(\mathcal{V}(t), \mathcal{E}(t), \mathbf{S}(t), \mathbf{x}(t)) . \tag{10f}$$

By fixing $h$ in the policy $\{h, \Psi\}$ to a given baseline heuristic $h'$ in (10d), we only need to optimize the actor network $\Psi(\cdot; \mathbf{\Theta}_p)$ that generates the intermediate action $\mathbf{c}(t)$ based on $\mathcal{G}(t)$ in (10e). Similar to (7), we estimate the gradient $\nabla_{\mathbf{c}(t)} \mathbb{E}_{\mathcal{G}(1) \sim \Omega^1} \left[ f_{obj} \left( \mathbf{o}(1) \right) \right]$ through a twin network that predicts the value vector in (10b) as $\mathbf{o}(t) \approx \hat{\mathbf{o}}(t) = f_{twin}(\mathcal{V}(t), \mathcal{E}(t), \mathbf{S}(t), \mathbf{c}(t); \mathbf{\Theta}_c)$. Based on linearity of expectation and the policy gradient theorem–the policy gradient does not depend on the gradient of the state distribution (Sutton et al., 1999), the policy gradient estimate is (Silver et al., 2014)

$$\widehat{\nabla_{\mathbf{\Theta}_p} \mathbb{E}}_{\mathcal{G}(1) \sim \Omega^1} \left[ f_{obj} \left( \mathbf{o}(1) \right) \right] \approx \mathbb{E}_{\mathcal{G}(1) \sim \Omega^1} \left[ \nabla_{\mathbf{\Theta}_p} \hat{\mathbf{o}}(t) \nabla_{\hat{\mathbf{o}}(t)} f_{obj}(\hat{\mathbf{o}}(1)) \right] . \tag{11}$$

Then, we can update the policy network as $\mathbf{\Theta}_p \leftarrow \mathbf{\Theta}_p + \alpha_p \nabla_{\mathbf{\Theta}_p} \hat{\mathbf{o}}(t) \nabla_{\hat{\mathbf{o}}(t)} f_{obj}(\hat{\mathbf{o}}(1))$ with $\mathcal{G}(1)$ sampled from $\Omega^1$. According to the derivation in Appendix C, the twin network can be trained via stochastic gradient descent, i.e., by minimizing the following loss with an off-the-shelf optimizer:

$$\ell_{mse}(\hat{\mathbf{o}}(t), \mathbf{r}(t) + \gamma \hat{\mathbf{o}}(t+1)) , \text{ where } \hat{\mathbf{o}}(t+1) = \mathbf{0}, \ \forall t \geq T . \tag{12}$$

## 4 NUMERICAL RESULTS

### 4.1 INDEPENDENT R-COPS

We demonstrate the effectiveness of GDPG-Twin on four types of independent R-COPs by showing that it can improve the quality of solutions of fast and/or distributed heuristics with minimal overhead. These problems are maximum weighted independent set (MWIS), minimum weighted dominating set (MWDS), node weighted Steiner tree (NWST), and minimum weighted connected dominating set (MWCDS). They are all NP-hard, and need to be solved repetitively in a wide range of applications. For example, the MWIS problem appears in various schedulers (Pinedo, 2012; Joo & Shroff, 2012; Zhao et al., 2022a) and multi-object tracking in computer vision (Brendel et al., 2011). The MWDS problem is encountered in wireless network clustering (Shahraki et al., 2020). Multicast routing in communication networks involves the NWST problem (Sun et al., 2020). The MWCDS problem can establish a virtual backbone network in wireless multihop networks, that covers all the nodes with

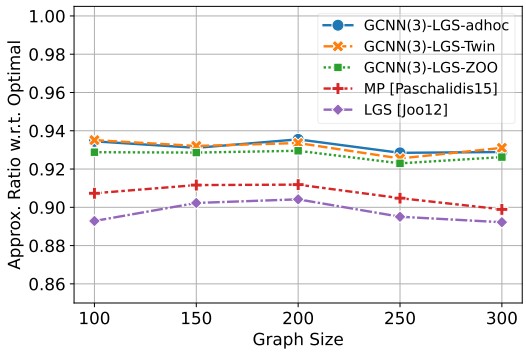

Figure 1: Approximation ratios (Larger is better) of the vanilla and GCNN-enhanced distributed heuristics for MWIS problem (max), w.r.t. the optimal solver.

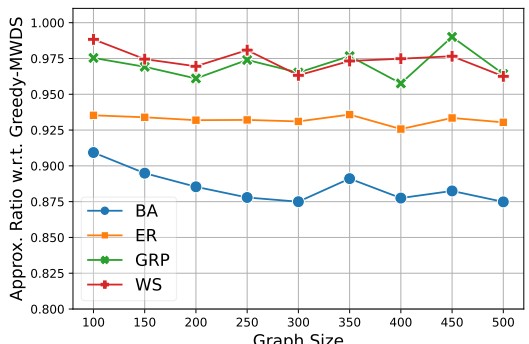

Figure 2: Approximation ratio (Smaller is better) of the GCNN-enhanced w.r.t. the vanilla Greedy-MWDS for MWDS problem (min) on 4 sets of random graphs.

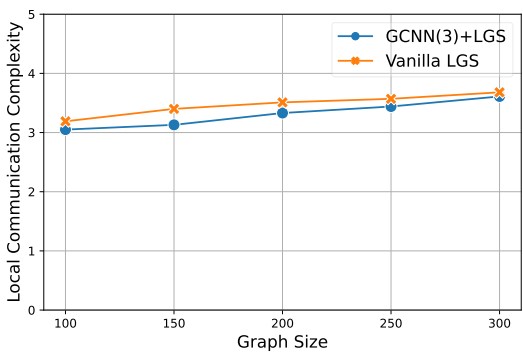

Figure 3: Average rounds of local message exchange for GCNN-enhanced and vanilla LGS-MWIS solvers to solve an instance, excluding the GCNN ($N = \infty$).

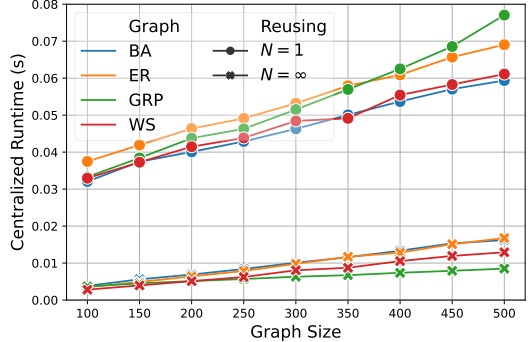

Figure 4: Average runtime of GCNN-enhanced Greedy-MWDS per instance by graph size, in seconds, no reusing $N = 1$ and infinity reusing $N = \infty$ of $\mathbf{Z}$.

the lowest cost in energy consumption or security vulnerability (Oliveira et al., 2011). We refer the readers to Appendix F for more detailed descriptions of the aforementioned applications.

We develop four similar ML pipelines respectively for the four R-COPs, in which the actor and twin networks are implemented by $L$-layer and 5-layer GCNNs, respectively, the baseline heuristics $h(\cdot)$ in (5c) are selected as centralized or distributed greedy solvers. The details of the four ML pipelines are in Appendix E for the equations of GCNN, Appendix F for the detailed training configurations, and Appendix H for the hyperparameters of GCNNs and brief descriptions of the chosen baseline heuristics. We aim to demonstrate the effectiveness of our proposed learning framework in closing the optimality gaps of some widely used fast heuristics for R-COPs, at the cost of negligible overhead, e.g., an additional computational complexity of $\mathcal{O}(L|\mathcal{E}|/N)$ or local communication complexity of $\mathcal{O}(L/N)$ per instance on sparse graphs. We do *not* claim smaller optimality gaps than the state-of-the-art (slower and centralized) heuristics, nor that GCNN is the best candidate graph neural architecture for these problems. An appealing characteristic of our framework is its modularity, which allows the GCNNs being replaced by any other (non-convolutional) GNN while our approach is still valid.

The four COPs are widely applied in various wireless networks, involving graphs of up to hundreds of nodes, and restricted runtimes from tens of milliseconds to a few seconds. Due to the dynamic nature and the lack of real-world datasets of wireless network topologies, we follow the norm of wireless research by running our experiments on four sets of synthetic random graphs: Erdős–Rényi (ER) (Erdős & Rényi, 1959), Barabási–Albert (BA) (Albert & Barabási, 2002), Gaussian Random Partition (GRP), and connected Watts-Strogatz small-world (WS) graphs. Each test dataset contains 2000 random graphs generated by a graph model with parameters detailed in Appendix F. The optimality of a heuristic is evaluated by the average approximation ratio, $\mathbb{E}_{\Omega'}(\mathbf{c}^{\top}\mathbf{x}^{(h)}/\mathbf{c}^{\top}\mathbf{x}^{(b)})$, on a test set $\Omega'$, where $\mathbf{x}^{(h)}$ and $\mathbf{x}^{(b)}$ are respectively the solutions from the heuristic of interests and a reference algorithm. Only in MWIS, the optimal solutions from (Zhao et al., 2022a) are used as reference, in other COPs, greedy heuristics are the references.

**MWIS:** An independent (vertex) set for a graph is a subset of vertices not connected by any edges. The MWIS problem is to find an independent set on a vertex weighted graph that maximizes the total weight. The baseline heuristic is a distributed local greedy solver (LGS-MWIS) (Joo & Shroff, 2012),

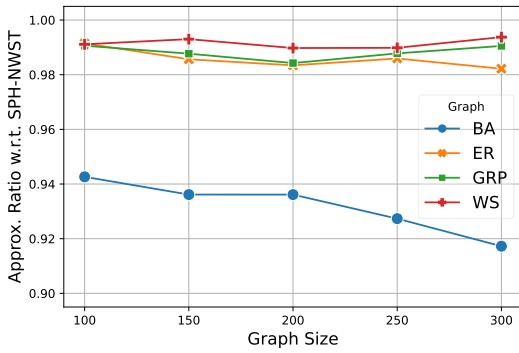 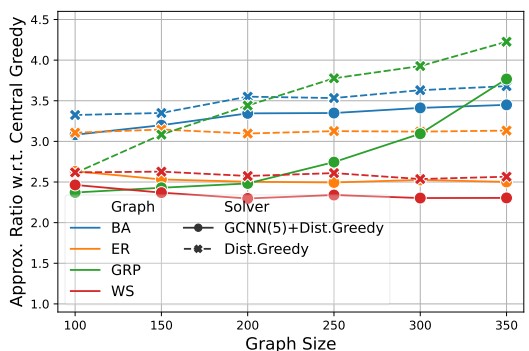

Figure 5: Approximation ratio (Smaller is better) of the GCNN-enhanced w.r.t. vanilla K-SPH-NWST for NWST problem on 4 sets of random graphs. NWST is a minimization (min) problem.

Figure 6: Approximation ratios (Smaller is better) of the vanilla and GCNN-enhanced distributed heuristics w.r.t. a centralized heuristic for MWCDS problem on 4 sets of random graphs. MWCDS is a min. problem.

with a local communication complexity of $\mathcal{O}(\log|\mathcal{V}|)$. We test multiple distributed MWIS solvers, including the GCNN-enhanced and vanilla LGS-MWIS, and a message passing (MP) algorithm (Paschalidis et al., 2015) on a set of 500 random ER graphs from (Zhao et al., 2022a). 3 GCNNs are respectively trained by ad hoc RL (Zhao et al., 2022a), GDPG-Twin, and ZOO (Liu et al., 2020), on a set of 6000 ER graphs. The approximation ratios of the tested solvers w.r.t. the optimal solver are shown in Figure 1. GDPG-Twin (93.1%) works equally well as the ad hoc RL (93.2%), and slightly outperforms ZOO (92.7%) with only ⅓ to ½ evaluations of $f_{net}(\cdot;\mathbf{Z})$ (see Section 4.3). The GCNN can reduce the optimality gap of LGS-MWIS (89.7%) by ⅓, beating the MP algorithm (90.7%). Figure 3 shows that GCNN-enhanced and vanilla LGS-MWIS converge in $3 \sim 4$ rounds in this test (MP algorithm converges in $2|\mathcal{V}|$ rounds), where the former is slightly faster for a large $N$.

**MWDS:** A dominating set for a graph $\mathcal{G} = (\mathcal{V}, \mathcal{E})$ is a subset $\mathcal{D}$ of $\mathcal{V}$ such that every vertex not in $\mathcal{D}$ is adjacent to at least one member of $\mathcal{D}$. In the MWDS problem, every node is associated with a non-negative weight, and the objective is to find a dominating set of minimum total weight. The baseline heuristic is a centralized greedy algorithm, Greedy-MWDS, as detailed in Appendix H. The approximation ratio of the GCNN-enhanced Greedy-MWDS w.r.t. the vanilla Greedy-MWDS on the 4 sets of random graphs described earlier, are shown in Figure 2. On average, GCNN improves the performance of Greedy-MWDS by 11.42% on BA graphs, 6.8% on ER graphs, 3.0% on GRP graphs, and 2.6% on WS graphs. The improvement is more pronounced on larger graphs. As shown in Figure 4, the average runtime of the GCNN-enhanced Greedy-MWDS on a laptop of 4 CPUs no GPU (detailed in Appendix I) ranges from 32 to 78 milliseconds per instance, and is linear to the graph size, which can be further cut to 2-17 milliseconds for a large reusing factor $N$.

**NWST:** In the NWST problem, we are given an undirected graph $\mathcal{G}$ with node cost (non-negative weight) and a subset of nodes called terminals. The goal is to find a minimum cost subgraph of $\mathcal{G}$ that connects the terminals. In the test, the terminals are selected by randomly removing $10\% \sim 50\%$ nodes from a maximal independent set (MIS) on the graph. The baseline heuristic is a distributed greedy algorithm, Kruskal's shortest path heuristic (K-SPH-NWST) (Matsuyama, 1980; Bauer & Varma, 1996). The approximation ratio of the GCNN-enhanced K-SPH-NWST w.r.t. the vanilla K-SPH-NWST on the 4 sets of random graphs, are shown in Figure 5. GCNN can improve K-SPH-NWST by 6.8% on BA graphs, 1.2% on GRP graphs, 1.4% on ER graphs, 0.9% on WS graphs, and 0.9% on a real-world dataset of Internet backbone topology (Knight et al., 2011) with $|\mathcal{V}| = 31.23$ on average. The noticeable gain on BA graphs is meaningful to large real-world networks, such as the Internet, World Wide Web, and social networks (Posfai & Barabasi, 2016, Ch. 5).

**MWCDS:** In MWCDS problem, we are given an undirected and connected graph, and our goal is to find a minimum weighted dominating set that is connected. Our baseline heuristic is a distributed greedy algorithm (Dist.Greedy), whereas the reference algorithm is a centralized greedy heuristic. Both MWCDS heuristics are implemented in two steps (Sun et al., 2019): 1) find a MWDS, 2) connect the MWDS by solving a NWST problem where the terminals are the solution of step 1. The approximation ratios of the vanilla and GCNN-enhanced distributed heuristics w.r.t. the centralized greedy heuristic on 4 sets of random graphs, each with 2000 graphs, are shown in Figure 6. Despite the large gap between distributed and centralized greedy algorithms, GCNN can improve the distributed greedy by 17.8% on GRP graphs, 16.0% on ER graphs, 6.6% on WS graphs, and 4.0% on BA graphs.

The average centralized runtimes of GCNN-enhanced distributed solvers for the MWIS, NWST, MWCDS problems are respectively $0.017 \sim 0.05$, $0.07 \sim 1.5$, $0.08 \sim 2.0$ seconds per instance, as

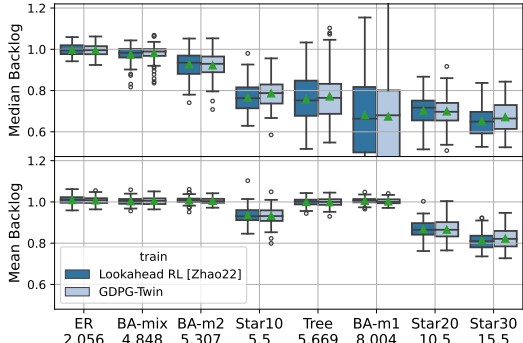 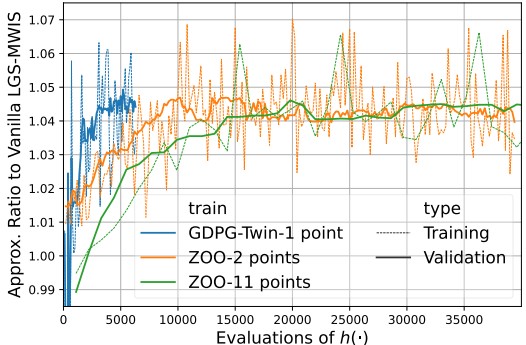

Figure 7: GDPG-Twin achieves similar network-wide mean and medium backlogs (smaller is better) of lookahead RL (Zhao et al., 2022b) in training a distributed link scheduler, using only ⅕ evaluations of $h(\cdot)$ of it.

Figure 8: Performance trajectories of GCNN-enhanced LGS-MWIS trained by GDPG-Twin and ZOOs with 2-point and 11-point gradient estimations. Larger is better. GDPG-Twin needs fewer evaluations of $h(\cdot)$.

reported in Appendix I, along with their estimated computational time (not the execution time) in distributed execution, as less than $0.18$, $5.1$, and $5.8$ milliseconds, respectively.

## 4.2 REPETITIVE MWIS IN A GRAPH-BASED MARKOV DECISION PROCESS

Next, we compare GDPG-Twin to an ad-hoc training method (Zhao et al., 2022b) in the context of delay-oriented distributed wireless link scheduling, demonstrating its effectiveness and efficiency in long-term goal-seeking for R-COPs in a graph-based MDP. The distributed scheduler contains a 1-layer GCNN and LGS in both methods, and a 5-layer GCNN is used as the twin in GDPG-Twin. The node feature matrix, $\mathbf{S}(t) = [\mathbf{q}(t); \mathbf{l}(t)]$, encloses the packet backlogs $\mathbf{q}(t)$ and stochastic data rates $\mathbf{l}(t)$ on all links (nodes in conflict graph). The state transition in (4f) is based on $\mathbf{q}(t+1) = \mathbf{q}(t) + \mathbf{a}(t) - \min(\mathbf{q}(t), \mathbf{l}(t) \odot \mathbf{x}(t))$, where $\mathbf{a}(t)$ captures random packets arrivals, and $\mathbf{x}(t)$ captures binary scheduling decisions. The objective is to minimize the average backlog, $\mathbb{E}_{i \in \mathcal{V}, t \leq T}[\mathbf{q}_i(t)]$, see Appendix A for detailed formulation. As shown in Figure 7, two distributed schedulers respectively trained by 5-step lookahead RL (Zhao et al., 2022b) and GDPG-Twin achieve similar performance in terms of the mean and median backlogs on conflict graphs with different levels of centralization, as indicated by the number on the x-axis. However, GPDG-Twin is 5 times faster than lookahead RL, as the former needs only 2 evaluations of $h(\cdot)$ per $t$, whereas the latter needs 10.

## 4.3 COMPARISON WITH ZEROTH-ORDER OPTIMIZATION IN TRAINING

The trajectories of relative performances of GCNN-enhanced LGS-MWIS (as detailed in Section 4.1) w.r.t. vanilla LGS-MWIS, under different training methods, are illustrated in Figure 8, where x-axis is the number of evaluations of the non-differentiable $h(\cdot)$ (LGS-MWIS). Each point on the curve is the average value of 100 instances. During training, random weights are generated on-the-fly for a training graph, whereas the weights on a validation graph are unchanged. GDPG-Twin and ZOO are configured with the same sampling radius and learning rate. In training, GDPG-Twin converges within only ⅓ to ½ evaluations of $h(\cdot)$ required by the fastest ZOO based on 2-point gradient estimation, showing a better sampling efficiency than ZOO, by a factor of 2 to 3.

## 5 CONCLUSION

We address repetitive combinatorial optimization problems under practical restrictions in runtime and/or distributed execution, by introducing a non-differentiable policy network based on a hand-picked, fast and/or distributed heuristic, which is parameterized by a continuous-valued high-dimensional intermediate action from an actor GNN. The actor GNN is optimized by graph-based deterministic policy gradient with the help of a critic based on a twin network that can predict the node-wise expected outcomes of the policy network. Through 5 examples, we demonstrate that our approach can: 1) leverage the shared underlying topology of independent R-COPs, to reduce the average optimality gap of the fast and/or distributed heuristics, and 2) optimize the long-term objectives for R-COPs in a graph-based Markov decision process. In terms of limitations, our work has not addressed the variance and worst-case of the optimality gap, which are important for many real-world applications. Our evaluation is based on synthetic rather than real-world graphs. Moreover, the actor and critic networks in our framework would need further design for broader tasks, e.g., edge related R-COPs. In addition, we do not expect our work to have any impact on social equality.

ACKNOWLEDGMENTS

This research was sponsored by the Army Research Office and was accomplished under Cooperative Agreement Number W911NF-19-2-0269. The views and conclusions contained in this document are those of the authors and should not be interpreted as representing the official policies, either expressed or implied, of the Army Research Office or the U.S. Government. The U.S. Government is authorized to reproduce and distribute reprints for Government purposes notwithstanding any copyright notation herein.

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

## A APPENDIX: TWO EXEMPLARY FORMULATIONS OF REPETITIVE COPS

For link scheduling in wireless multiple networks (Joo & Shroff, 2012; Zhao et al., 2022a;b), the network state is $\mathcal{G}(t) = (\mathcal{V}(t), \mathcal{E}(t), \mathbf{S}(t))$, where $(\mathcal{V}(t), \mathcal{E}(t))$ is the underlying topology of the conflict graph, in which a vertex $i \in \mathcal{V}(t)$ represents a wireless link, and an undirected edge $\{i, j\} \in \mathcal{E}(t)$ captures the conflicting relationship between two links $i$ and $j$. The underlying topology of the conflict graph is supposed to change slowly compared to node features $\mathbf{S}(t)$.

The node feature matrix $\mathbf{S}(t) = [\mathbf{q}(t); \mathbf{l}(t)]$ encloses vectors of packet backlogs (queue lengths), $\mathbf{q}(t)$, and predicted link rates $\mathbf{l}(t)$ on all the links at time $t$. The backlog vector $\mathbf{q}(t)$ evolves according to the scheduling decisions $\mathbf{x}(t)$, and the state transition function $f_s(\cdot)$ in (4f) is

$$\mathbf{q}(t+1) = \mathbf{q}(t) + \mathbf{a}(t) - \min(\mathbf{q}(t), \mathbf{l}(t) \odot \mathbf{x}(t)),$$

where $\mathbf{a}(t)$ captures random packets arrivals, and $\mathbf{l}(t)$ follows a stationary random distribution.

The scheduling decision is subjective to a binary constraint, $\mathbf{x}_i \in \{0, 1\}, \forall i \in \mathcal{V}$, and a constraint of independent set as, $\mathbf{x}_i + \mathbf{x}_j \leq 1, \forall \{i, j\} \in \mathcal{E}$, corresponding to (2b) and (2c), respectively.

In link scheduling for throughput maximization (Joo & Shroff, 2012; Zhao et al., 2022a), (4e) is defined as utility function $\mathbf{c}(t) = \mathbf{q}(t) \odot \mathbf{l}(t)$ to prioritize wireless links with large backlogs and high link rates. The heuristic $h(\cdot)$ in (4d) is a distributed MWIS solver, e.g., LGS in (Joo & Shroff, 2012) and GCN-LGS in (Zhao et al., 2022a). The reward vector in (4c) is defined as $\mathbf{r}(t) = \mathbf{c}(t) \odot \mathbf{x}(t)$, and the expected return $\mathbf{o}(t) = \mathbb{E}_{t \leq T}[\mathbf{r}(t)]$ for (4b). Since throughput maximization is equivalent to optimize MWIS instances individually without considering their inter-dependencies, (4e) and (4f) together can be simplified as a sampling process $\mathbf{c} \sim \Omega^\mathbf{c}$. Therefore, $t$ can be dropped, and the expected return becomes $\mathbf{o}(t) = \mathbb{E}_{\mathbf{c} \sim \Omega^\mathbf{c}}[\mathbf{c} \odot \mathbf{x}]$, and the objective function in (4a) becomes $f_{obj}(\mathbf{o}(t)) = \mathbf{1}^\top \mathbf{o} = \mathbb{E}_{\mathbf{c} \sim \Omega^\mathbf{c}}(\mathbf{c}^\top \mathbf{x})$. The formulation is then simplified as (3) with $\mathbf{c} = -\mathbf{q} \odot \mathbf{l}$.

For delay-oriented link scheduling (Zhao et al., 2022b), the objective is to minimize the average backlog across the network and time, $\mathbb{E}_{i \in \mathcal{V}, t \leq T}[\mathbf{q}_i(t)]$, and the inter-dependencies between network states cannot be ignored. This objective can be formulated as an average reward objective, implemented by setting the discount factor as $\gamma = 1$ and defining $f_{obj}(\mathbf{o}(t)) = \frac{1}{T}\mathbb{E}_{i \in \mathcal{V}}[\mathbf{o}_i(t)]$. The heuristic solver $h(\cdot)$ in (4d) is LGS, and the reward vector in (4c) is $\mathbf{r}(t) = \mathbf{q}'(t) - \mathbf{q}(t)$, where $\mathbf{q}'(t)$ is the backlog vector under LGS with a baseline cost vector $\mathbf{c}'(t-1) = \mathbf{q}(t-1) \odot \mathbf{l}(t-1)$. The objective function in (4a) can then be transformed to $\mathbb{E}_{i \in \mathcal{V}, t \leq T}[\mathbf{q}'_i(t)] - \mathbb{E}_{i \in \mathcal{V}, t \leq T}[\mathbf{q}_i(t)]$, in which the first component is a constant. The cost vector in (4e) is obtained by a GCN as $\mathbf{c}(t) = \Psi(\mathcal{V}(t), \mathcal{E}(t), \mathbf{q}(t) \odot \mathbf{l}(t); \; \boldsymbol{\Theta}_p)$, which is trained to maximize the objective function.

These two examples show that independent R-COP is a special case of R-COP in graph-based MDP, and problems under similar settings can be formulated differently based on their objectives.

## B APPENDIX: SGD FOR TWIN NETWORK IN INDEPENDENT REPETITIVE COPS

Give MSE loss,

$$\ell_{mse}(\hat{\mathbf{o}}, \mathbf{o}) = \frac{1}{|\mathbf{o}|} \sum_{i=1}^{|\mathbf{o}|} (\hat{\mathbf{o}}_i - \mathbf{o}_i)^2 , \tag{13}$$

the partial derivative of $\ell_{mse}(\hat{\mathbf{o}}, \mathbf{o})$ w.r.t. the parameters of the twin network is

$$\frac{\partial \ell_{mse}(\hat{\mathbf{o}}, \mathbf{o})}{\partial \boldsymbol{\Theta}_c} = \frac{\partial \ell_{mse}(\hat{\mathbf{o}}, \mathbf{o})}{\partial \hat{\mathbf{o}}} \frac{\partial \hat{\mathbf{o}}}{\partial \boldsymbol{\Theta}_c} \tag{14a}$$

$$= \frac{2}{|\mathbf{o}|} [\hat{\mathbf{o}} - \mathbb{E}_{\Omega^{|\mathbf{c}|}}(\mathbf{c} \odot \mathbf{x})]^\top \frac{\partial \hat{\mathbf{o}}}{\partial \boldsymbol{\Theta}_c} \tag{14b}$$

$$= \mathbb{E}_{\Omega^{|\mathbf{c}|}} \left( \frac{2}{|\mathbf{o}|} (\hat{\mathbf{o}} - \mathbf{c} \odot \mathbf{x})^\top \frac{\partial \hat{\mathbf{o}}}{\partial \boldsymbol{\Theta}_c} \right) , \tag{14c}$$

where from (14b) to (14c) is based on the linearity of expectation. Based on (14c), by drawing $\mathbf{c} \sim \Omega^{\mathbf{c}}$, we have the following unbiased stochastic gradient estimation:

$$\frac{\partial \ell_{mse}(\hat{\mathbf{o}}, \mathbf{o})}{\partial \mathbf{\Theta}_c} = \mathbb{E}_{\Omega^{\mathbf{c}}} \left[ \frac{\widehat{\partial \ell_{mse}(\hat{\mathbf{o}}, \mathbf{o})}}{\partial \mathbf{\Theta}_c} \right] , \tag{15}$$

where

$$\frac{\widehat{\partial \ell_{mse}(\hat{\mathbf{o}}, \mathbf{o})}}{\partial \mathbf{\Theta}_c} = \frac{2}{|\mathbf{o}|} (\hat{\mathbf{o}} - \mathbf{c} \odot \mathbf{x})^\top \frac{\partial \hat{\mathbf{o}}}{\partial \mathbf{\Theta}_c} \tag{16a}$$

$$= \frac{\partial \ell_{mse}(\hat{\mathbf{o}}, \mathbf{c} \odot \mathbf{x})}{\partial \mathbf{\Theta}_c} . \tag{16b}$$

Therefore, the stochastic gradient estimation for the parameters of the twin network is

$$\widehat{\nabla_{\mathbf{\Theta}_c} \ell_{mse}}(\hat{\mathbf{o}}, \mathbf{o}) = \nabla_{\mathbf{\Theta}_c} \ell_{mse}(\hat{\mathbf{o}}, \mathbf{c} \odot \mathbf{x}) . \tag{17}$$

## C  APPENDIX: SGD FOR TWIN NETWORK IN R-COPS IN GRAPH-BASED MDP

We define a return vector supported on the nodes of the graph as

$$\mathbf{g}(t) = \sum_{k=0}^{T-t} \gamma^k \mathbf{r}(t+k) , \tag{18}$$

which can be expressed in a recursive form

$$\mathbf{g}(t) = \mathbf{r}(t) + \gamma \mathbf{g}(t+1) , \text{ where } \mathbf{g}(t) = \mathbf{0} , \forall t > T . \tag{19}$$

The outcome (value) vector of state $\mathcal{G}(t)$ is the expected return vector under policy $\mathbf{\Theta}_p$,

$$\mathbf{o}(t) = \mathbb{E}_{\mathbf{\Theta}_p} [\mathbf{g}(t)|\mathcal{G}(t)] . \tag{20}$$

The gradient $\nabla_{\mathbf{\Theta}_c} \ell_{mse}(\hat{\mathbf{o}}(t), \mathbf{o}(t))$ can be found by the following partial derivative

$$\frac{\partial \ell_{mse}(\hat{\mathbf{o}}(t), \mathbf{o}(t))}{\partial \mathbf{\Theta}_c} = \frac{\partial \ell_{mse}(\hat{\mathbf{o}}(t), \mathbf{o}(t))}{\partial \hat{\mathbf{o}}(t)} \frac{\partial \hat{\mathbf{o}}(t)}{\partial \mathbf{\Theta}_c} \tag{21a}$$

$$= \frac{2}{|\mathbf{o}(t)|} \left( \hat{\mathbf{o}}(t) - \mathbb{E}_{\mathbf{\Theta}_p} [\mathbf{g}(t)|\mathcal{G}(t)] \right)^\top \frac{\partial \hat{\mathbf{o}}(t)}{\partial \mathbf{\Theta}_c} \tag{21b}$$

$$= \mathbb{E}_{\mathbf{\Theta}_p} \left( \frac{2}{|\mathbf{o}(t)|} [\hat{\mathbf{o}}(t) - \mathbf{g}(t)|\mathcal{G}(t)]^\top \frac{\partial \hat{\mathbf{o}}(t)}{\partial \mathbf{\Theta}_c} \right) , \tag{21c}$$

where from (21b) to (21c) is based on the linearity of expectation, and that $\hat{\mathbf{o}}(t)$ is a function of $\mathcal{G}(t)$, generated by the twin network, as defined in (12). Based on (19), we have the following unbiased estimation of the partial derivative in (21a)

$$\frac{\partial \ell_{mse}(\hat{\mathbf{o}}(t), \mathbf{o}(t))}{\partial \mathbf{\Theta}_c} = \mathbb{E}_{\mathbf{\Theta}_p} \left[ \frac{\widehat{\partial \ell_{mse}(\hat{\mathbf{o}}(t), \mathbf{o}(t))}}{\partial \mathbf{\Theta}_c} \right] , \tag{22}$$

where

$$\frac{\widehat{\partial \ell_{mse}(\hat{\mathbf{o}}(t), \mathbf{o}(t))}}{\partial \mathbf{\Theta}_c} = \frac{2}{|\mathbf{o}(t)|} [\hat{\mathbf{o}}(t) - \mathbf{g}(t)|\mathcal{G}(t)]^\top \frac{\partial \hat{\mathbf{o}}(t)}{\partial \mathbf{\Theta}_c} \tag{23a}$$

$$= \frac{2}{|\mathbf{o}(t)|} [\hat{\mathbf{o}}(t) - \mathbf{r}(t) - \gamma \mathbf{g}(t+1)|\mathcal{G}(t)]^\top \frac{\partial \hat{\mathbf{o}}(t)}{\partial \mathbf{\Theta}_c} . \tag{23b}$$

In (23b), $\mathbf{r}(t)$ is the reward vector collected under current state $\mathcal{G}(t)$, as defined in (10c), and $\mathbf{g}(t+1)|\mathcal{G}(t)$ is the return of the next state $\mathcal{G}(t+1)$, evolved from the current state $\mathcal{G}(t)$ according to (10f). The return of the next state $\mathcal{G}(t+1)$ is estimated as $\mathbf{g}(t+1) \approx \hat{\mathbf{o}}(t+1)$ by the twin network, as defined in (12). As a result, we have the following approximation

$$\mathbf{r}(t) + \gamma \mathbf{g}(t+1)|\mathcal{G}(t) \approx \mathbf{r}(t) + \gamma \hat{\mathbf{o}}(t+1) . \tag{24}$$

Plugging (24) to (23b), we have

$$\frac{\widehat{\partial \ell_{mse}(\hat{\mathbf{o}}(t), \mathbf{o}(t))}}{\partial \mathbf{\Theta}_c} \approx \frac{\partial \ell_{mse}(\hat{\mathbf{o}}(t), \mathbf{r}(t) + \gamma \hat{\mathbf{o}}(t+1))}{\partial \mathbf{\Theta}_c} . \tag{25}$$

Therefore, the stochastic gradient estimation for the parameters of the twin network is

$$\widehat{\nabla_{\mathbf{\Theta}_c} \ell_{mse}}(\hat{\mathbf{o}}(t), \mathbf{o}(t)) \approx \nabla_{\mathbf{\Theta}_c} \ell_{mse}(\hat{\mathbf{o}}(t), \mathbf{r}(t) + \gamma \hat{\mathbf{o}}(t+1))) . \tag{26}$$

# D  APPENDIX: ALGORITHMIC PROCEDURES OF GDPG-TWIN

---

**Algorithm 1** GDPG-Twin for R-COPs with Independent Instances

---

   **Input**: $\Omega^{\mathbf{c}}, \Omega^{\mathcal{G}}, h(\cdot), \alpha_p, \alpha_c, E, B, \epsilon$
   **Output**: $\mathbf{Z}$ or $\boldsymbol{\Theta}_p, \boldsymbol{\Theta}_c$
1: Initialize $\mathbf{Z}$ or $\boldsymbol{\Theta}_p, \boldsymbol{\Theta}_c$ randomly or as pretrained models, $\bar{\mathbf{c}} = \mathbb{E}_{\Omega^{\mathbf{c}}}(\mathbf{c})$
2: **for** $e \in \{1, 2, \ldots, E\}$ **do**
3:    $\mathcal{Q}_p = \varnothing, \mathcal{Q}_c = \varnothing$ */* Clear gradient buffers */
4:    **for** $b \in \{1, \ldots, B\}$ **do**
5:       Draw $\mathcal{G}(\mathcal{V}, \mathcal{E}) \in \Omega^{\mathcal{G}}, \mathbf{c} \in \Omega^{\mathbf{c}}$ */* Draw data from training dataset or target distribution */
6:       **if** Actor network is used **then**
7:          $\mathbf{Z} = \Psi(\mathcal{V}, \mathcal{E}, \bar{\mathbf{c}}; \boldsymbol{\Theta}_p)$
8:       **end if**
9:       $\mathbf{Z}^{(j)} = \mathbf{Z} + \mathbf{N}^{(j)}, \mathbf{N}^{(j)} \in \mathbb{U}(-\epsilon, \epsilon)$ */* Random policy sampling */
10:      $\hat{\mathbf{o}} = f_{twin}(\mathcal{V}, \mathcal{E}, \bar{\mathbf{c}}, \mathbf{Z}^{(j)}; \boldsymbol{\Theta}_c^*)$
11:      $\mathbf{x} = f_{net}(\mathcal{V}, \mathcal{E}, \mathbf{c}; \mathbf{Z}^{(j)})$ based on $h(\cdot)$ in (5c) and $f_{loc}(\cdot)$ in (5d)
12:      Estimate gradient $\nabla_{\boldsymbol{\Theta}_c} \ell_{mse}(\hat{\mathbf{o}}, \mathbf{c} \odot \mathbf{x})$ for critic
13:      $\mathcal{Q}_c \leftarrow \mathcal{Q}_c \cup \{\nabla_{\boldsymbol{\Theta}_c} \ell_{mse}(\hat{\mathbf{o}}, \mathbf{c} \odot \mathbf{x})\}$
14:      Estimate policy gradient $\nabla_{\mathbf{Z}} \mathbf{1}^\top \hat{\mathbf{o}}$ based on (7)
15:      **if** Actor network is used **then**
16:         Estimate gradient $\nabla_{\boldsymbol{\Theta}_p} \mathbf{1}^\top \hat{\mathbf{o}} = \nabla_{\boldsymbol{\Theta}_p} \Psi(\mathcal{V}, \mathcal{E}, \bar{\mathbf{c}}; \boldsymbol{\Theta}_p) \nabla_{\mathbf{Z}} \mathbf{1}^\top \hat{\mathbf{o}}$ for actor
17:         $\mathcal{Q}_p \leftarrow \mathcal{Q}_p \cup \{\nabla_{\boldsymbol{\Theta}_p} \mathbf{1}^\top \hat{\mathbf{o}}\}$
18:      **else**
19:         $\mathcal{Q}_p \leftarrow \mathcal{Q}_p \cup \{\nabla_{\mathbf{Z}} \mathbf{1}^\top \hat{\mathbf{o}}\}$
20:      **end if**
21:    **end for**
22:    $\boldsymbol{\Theta}_c \leftarrow \boldsymbol{\Theta}_c - \alpha_c \mathbb{E}_{\mathcal{Q}_c} \left[ \nabla_{\boldsymbol{\Theta}_c} \ell_{mse}(\hat{\mathbf{o}}, \mathbf{c} \odot \mathbf{x}) \right]$
23:    $\mathbf{Z} \leftarrow \mathbf{Z} - \alpha_p \mathbb{E}_{\mathcal{Q}_c} \left( \nabla_{\mathbf{Z}} \mathbf{1}^\top \hat{\mathbf{o}} \right)$ or $\boldsymbol{\Theta}_p \leftarrow \boldsymbol{\Theta}_p - \alpha_p \mathbb{E}_{\mathcal{Q}_p} \left( \nabla_{\boldsymbol{\Theta}_p} \mathbf{1}^\top \hat{\mathbf{o}} \right)$
24: **end for**
25: Output $\mathbf{Z}$ or $\boldsymbol{\Theta}_p, \boldsymbol{\Theta}_c$

---

For the four demonstrated R-COPs with independent instances, we set the hyperparameters of training procedure in Algorithm 1 as follows: $\alpha_p = \alpha_c = 0.0001$, $E = 25$, $B = 100$, $\epsilon = 0.15$.

---

**Algorithm 2** GDPG-Twin for R-COPs in a graph-based MDP

---
  **Input**: $\Omega^1, \Omega^{\mathcal{G}}, h(\cdot), \alpha_p, \alpha_c, T, E$
  **Output**: $\boldsymbol{\Theta}_p, \boldsymbol{\Theta}_c$
 1: Initialize $\boldsymbol{\Theta}_p, \boldsymbol{\Theta}_c$ randomly or as pretrained models
 2: **for** $e \in \{1, 2, \dots, E\}$ **do**
 3:     $\mathcal{Q}_e = \varnothing, \mathcal{Q}_p = \varnothing, \mathcal{Q}_c = \varnothing$ /* Clear experience & gradient buffers */
 4:     Initialize state $\mathcal{G}(1) = (\mathcal{V}(1), \mathcal{E}(1), \mathbf{S}(1)) \sim \Omega^1$ (or $\Omega^{\mathcal{G}}$ for ergodic MDP)
 5:     **for** $t \in \{1, \dots, T\}$ **do**
 6:         $\mathbf{c}(t) = \Psi(\mathcal{V}(t), \mathcal{E}(t), \mathbf{S}(t); \boldsymbol{\Theta}_p)$
 7:         $\hat{\mathbf{o}}(t) = f_{twin}(\mathcal{V}(t), \mathcal{E}(t), \mathbf{S}(t), \mathbf{c}(t); \boldsymbol{\Theta}_c^*)$
 8:         Obtain decision vector $\mathbf{x}(t)$ based on (10d)
 9:         Observe reward vector $\mathbf{r}(t)$ according to (10c)
10:         Update state feature $\mathbf{S}(t + 1)$ according to (10f)
11:         Estimate stochastic policy gradient $\nabla_{\boldsymbol{\Theta}_p} f_{obj}(\hat{\mathbf{o}}(t))$ based on (11)
12:         $\mathcal{Q}_p \leftarrow \mathcal{Q}_p \cup \{\nabla_{\boldsymbol{\Theta}_p} f_{obj}(\hat{\mathbf{o}}(t))\}$
13:         $\mathcal{Q}_e \leftarrow \mathcal{Q}_e \cup \{< \mathbf{r}(t), \hat{\mathbf{o}}(t) >\}$
14:     **end for**
15:     **for** $t \in \{1, \dots, T\}$ **do**
16:         Fetch $\mathbf{r}(t), \hat{\mathbf{o}}(t), \hat{\mathbf{o}}(t + 1)$ from $\mathcal{Q}_e$
17:         Estimate stochastic gradient $\nabla_{\boldsymbol{\Theta}_c} \ell_{mse}(\hat{\mathbf{o}}(t), \mathbf{r}(t) + \gamma \hat{\mathbf{o}}(t + 1))$ for critic
18:         $\mathcal{Q}_c \leftarrow \mathcal{Q}_c \cup \{\nabla_{\boldsymbol{\Theta}_c} \ell_{mse}(\hat{\mathbf{o}}(t), \mathbf{r}(t) + \gamma \hat{\mathbf{o}}(t + 1))\}$
19:     **end for**
20:     $\boldsymbol{\Theta}_c \leftarrow \boldsymbol{\Theta}_c - \alpha_c \mathbb{E}_{\mathcal{Q}_c} [\nabla_{\boldsymbol{\Theta}_c} \ell_{mse}(\hat{\mathbf{o}}(t), \mathbf{r}(t) + \gamma \hat{\mathbf{o}}(t + 1))]$
21:     $\boldsymbol{\Theta}_p \leftarrow \boldsymbol{\Theta}_p + \alpha_p \nabla_{\boldsymbol{\Theta}_p} f_{obj}(\hat{\mathbf{o}}(1))$ /* Objective maximization */
22: **end for**
23: Output $\boldsymbol{\Theta}_p, \boldsymbol{\Theta}_c$

---

For the demonstrated delay-oriented link scheduling (Zhao et al., 2022b), we use the following hyperparameters in training described in Algorithm 2: $\alpha_p = \alpha_c = 0.0001$, $T = 64$, $E = 200$. The types and mixture of the training graphs, random processes, etc., are identical to (Zhao et al., 2022b).

## E  APPENDIX: GRAPH CONVOLUTIONAL NEURAL NETWORKS

An $L$-layer GCNN is implemented as follows: Given the input feature $\mathbf{S}^{(0)} = \mathbf{S}$ supported on a graph $\mathcal{G}$, the output is $\mathbf{Z} = \mathbf{S}^{(L)} = f_{GCN}(\mathcal{G}, \mathbf{S}; \boldsymbol{\Theta})$, where an intermediate $l$th layer of the GCNN is

$$\mathbf{S}^l = \sigma_l \left( \mathbf{S}^{l-1} \boldsymbol{\Theta}_0^l + \boldsymbol{\mathcal{L}} \mathbf{S}^{l-1} \boldsymbol{\Theta}_1^l \right) , \; l \in \{1, \dots, L\} . \tag{27}$$

In (27), $\boldsymbol{\mathcal{L}}$ is the normalized Laplacian of graph $\mathcal{G}$, $\boldsymbol{\Theta}_0^l, \boldsymbol{\Theta}_1^l \in \mathbb{R}^{g_{l-1} \times g_l}$ are the trainable parameters, $g_{l-1}$ and $g_l$ are the dimensions of the output features of layers $l - 1$ and $l$, respectively, and $\sigma_l(.)$ is an element-wise activation function.

In a distributed system, (27) can be implemented by the following local operation on node $v \in \mathcal{V}$,

$$\mathbf{S}_{v*}^l = \sigma_l \left( \mathbf{S}_{v*}^{l-1} \boldsymbol{\Theta}_0^l + \left[ \mathbf{S}_{v*}^{l-1} - \sum_{u \in \mathcal{N}(v)} \frac{\mathbf{S}_{u*}^{l-1}}{\sqrt{d(v)d(u)}} \right] \boldsymbol{\Theta}_1^l \right) , \tag{28}$$

where $\mathbf{S}_{v*}^l \in \mathbb{R}^{1 \times g_l}$ is the $v$th row of matrix $\mathbf{S}^l$, which captures the features on node $v$, $d(v)$ is the degree of node $v$, and $\mathcal{N}(v)$ is the set of neighboring nodes of node $v$.

## F  APPENDIX: CONFIGURATIONS OF RANDOM GRAPHS FOR TRAINING AND TESTING

$p \in \mathbb{U}(0.15, 0.35)$, $k \in \lfloor \mathbb{U}(10, 30) \rceil$. Training graph size $|\mathcal{V}| \in \{100, 150, 200, 250, 300\}$. Testing graph size $|\mathcal{V}| \in \{100, 150, 200, 250, 300, 350, 400, 450, 500\}$ (varies by problem).

ER: size $|\mathcal{V}|$, edge probability $|\mathcal{V}|/k$. BA: size $|\mathcal{V}|$, number of edges to attach from a new node to existing nodes $m = \lfloor pk \rfloor$. WS: size $|\mathcal{V}|$, each node is joined with its $k$ nearest neighbors in a ring topology, $p$: the probability of rewiring each edge. GRP: size $|\mathcal{V}|$, mean cluster size $k$, shape parameter $\min(7, k)$, probability of intra-cluster connection $p$, probability of inter cluster connection $\max(0.1, p/3)$. For MWIS problem in Section 4.1, the test set of 500 ER graphs and the corresponding optimal solutions are from `https://github.com/zhongyuanzhao/distgcn` (Zhao et al., 2022a).

# G  APPENDIX: APPLICATIONS OF THE FOUR EXEMPLARY R-COPS

## G.1  MWIS IN SCHEDULING AND COMPUTER VISION

**Definition:** An independent (vertex) set for a graph is a subset of vertices not connected by any edges. The MWIS problem is to find an independent set on a vertex weighted graph that maximizes the total weight.

**MWIS for wireless scheduling:** MWIS can be applied to link scheduling in wireless multihop networks with orthogonal multiple access (Joo & Shroff, 2012; Zhao et al., 2022a). In a wireless multihop network, a wireless link refers to a pair of nearby wireless transceivers that can directly talk to each other. In orthogonal multiple access, two links would conflict with each other if they share the same transceiver (which can only be tuned to one link at a time), or they would block out each other if activated simultaneously (e.g., any transceiver(s) of a link are located too close to any transceiver(s) of the other link to interfere the reception of wireless signal). In each time slot, a link scheduler (Max-Weight scheduler) would determine a set of non-interfering links to be activated, so that it would maximize the total utility of the wireless network. Max-Weight scheduling is essentially finding a MWIS on the conflict graph of the wireless network, which is defined as follows: each vertex in the conflict graph is a link in the wireless network, and an edge captures the conflict relationship between two links. For example, with a per-link utility function based on the length of backlogged data packets of each link, Max-Weight scheduling can achieve the maximum throughput (the amount of data packets transmitted in a time slot) of the wireless network. The typical length of a time slot in various wireless communication protocols ranges from $1 \sim 100$ milliseconds, which means that a Max-Weight scheduler needs to solve an MWIS instance every $1 \sim 100$ milliseconds. Meanwhile, the topology of the conflict graph in wireless scheduling (determined by the topology of the wireless networks and physical locations of transceivers) evolves at much lower pace, such as seconds to minutes for mobile wireless networks, or remain the same if all the transceivers (such as microwave towers and wireless sensors) are static.

Graph coloring problems are also applied to wireless scheduling, especially, multi-channel wireless scheduling. The conflict graph is formulated similarly in the previous single-channel scheduling, but instead of finding a MWIS, each node in the conflict graph (link in the wireless network) is assigned a color representing a particular channel, so that neighboring nodes (conflicting links) will never have the same color (channel). In Cornaz et al. (2017), the four types of graph coloring problems: Vertex coloring problem (VCP), equitable vertex coloring problem (ECP), Max-coloring (Max-Col) which can be seen as the weighted version of VCP, and Bin Packing Problem with Conflict (BPPC), can be converted to solving MWIS on an associated graph.

**MWIS for multiobject tracking in computer vision:**  In multiobject tracking (Brendel et al., 2011), a detector first identifies a set of objects in a video frame, records the following properties of the corresponding bounding box of each object: location, size, the histograms of color, intensity gradients, and optical flow. Next, the detected objects across different video frames needs to be linked according to their properties to maintain their unique identities. The second step is formulated as finding a MWIS on a graph, in which nodes represent candidate matches (of two objects) from every two consecutive frames, referred as tracklets; node weights encode the similarity of the corresponding matches; and edges connect nodes whose corresponding tracklets violate the hard constraints that no two matches share the same object. If there are 10 objects in each of the two consecutive frames, there would be 100 tracklets. The MWIS on such a graph is a set of matches that maximize the total similarities of tracklets, resulting in the most plausible tracking of multiple objects. In this formulation, a MWIS instance needs to be solved every video frame (24, 30 frames per second), while the topology of the graphs of consecutive MWIS instances would be similar since consecutive frames in a typical video stream would remain similar.

## G.2 MWDS IN WIRELESS SENSOR NETWORKS AND COVERING CODES

**Definition:** A dominating set for a graph $\mathcal{G} = (\mathcal{V}, \mathcal{E})$ is a subset $\mathcal{D}$ of $\mathcal{V}$ such that every vertex not in $\mathcal{D}$ is adjacent to at least one member of $\mathcal{D}$. In the MWDS problem, every node is associated with a non-negative weight, and the objective is to find a dominating set of minimum total weight.

**MWDS for clustering in wireless sensor networks:** Wireless sensor networks are a type of ad-hoc network for monitoring purposes. It usually include a large number of sensor nodes, which are resource-constrained (such as battery power), but can connect to other nodes of the network for transmitting sensed data. Each node can also forward data from neighbors to the sink (or gateway, base station, server, etc.) (Shahraki et al., 2020). Clustering is one of the most popular techniques for the topology management of wireless sensor networks . It organizes sensor nodes into a set of groups called clusters, each cluster has one or more cluster heads which gathers data from other members of the cluster and send the (fused) data to the sink directly or indirectly. Using clustering techniques, resource-constrained sensor nodes do not need to send their data to the sink directly, which can cause energy depletion, resource consumption inefficiency and interference. Clustering can be formulated as MWDS problem, where the cluster heads form a dominating set so that the rest of the sensor nodes can directly reach at least one cluster head. By defining the node weight as the cost of being a cluster head, MWDS can minimize the total cost of wireless sensor networks, such as energy consumption or quality of service. To maximize the lifetime of the wireless sensor network, it may need to select a different set of cluster heads once in a while to avoid draining the battery of cluster heads. The node weight would change based on the battery levels of the sensor nodes.

## G.3 NWST FOR MULTICAST ROUTING IN WIRELESS NETWORKS

**Definition:** In the NWST problem, we are given an undirected graph $\mathcal{G}$ with node cost (non-negative weight) and a subset of nodes called terminals. The goal is to find a minimum cost subgraph of $\mathcal{G}$ that connects the terminals.

**NWST in multicast routing:** A multicast route is a network route that connects more than two nodes at the same time (Oliveira et al., 2011). Multicast routing is applicable to networking scenarios where data needs to be shared by a group of users, such as a software company sends out a security patch to the computers across the Internet installed with its software, or a company pushes a notification to the smart phones installed with its mobile app. In these example, the Internet would be the graph, the server and the recipients are defined as the terminals in the Steiner tree problem, and the non-terminal nodes are the routers, gateways, and other computers on the Internet.

Multicast routing in wireless multihop networks can be formulated as node weighted Steiner tree (NWST) problem (Sun et al., 2020). The graph models the network, e.g., a node represents a wireless device and an edge represents a link between two wireless devices. The terminals in the NWST are the subset of wireless devices that need to connected with each other, and the non-terminal nodes are the rest of the wireless devices in the network. The node weight captures the cost of a non-terminal node relaying packets, such as the consumption of energy, bandwidth, and/or CPU load. By formulating the multicast routing as a NWST, we can connect all the terminals at minimal total cost. Whenever there is a request to establish a multicast route, a NWST problem in the network needs to be solved. In medium to large-scale wireless networks, distributed algorithms are always preferred since it does not require a centralized coordinator to collect the real-time network state and compute the solution.

## G.4 MWCDS IN MOBILE AD-HOC NETWORKS

**Definition:** In MWCDS problem, we are given an undirected connected graph, and our goal is to find a minimum weighted dominating set that is connected.

**MWCDS for virtual backbone computation:** In mobile ad-hoc networks, a virtual backbone is a set of nodes that can be used to take routing decisions and that act as proxies for routing packets (Oliveira et al., 2011). It can reduce the amount of routing information shared among nodes of the system, by requiring only a subset of nodes to be actively involved in routing decisions. Formally, we would define the mobile ad-hoc network as a undirected graph, in which each node represents a mobile device, and each edge represents a link between two mobile devices. A virtual backbone needs to a connected subgraph so that packets from one node can always reach to another node in the virtual backbone. A node in the network needs to be either a member of the virtual backbone

or adjacent to at least one member of the virtual backbone, so that it can reach to the rest of the network. Therefore, a virtual backbone needs to be a dominating set of the graph. Moreover, since members of the virtual backbone take the responsibility of routing, which comes with extra cost in battery power consumption and routing packets, we want to minimize the total cost of implementing a virtual backbone. Therefore, the computation of a virtual backbone is to find the minimum weighted connected dominating set of the network. The virtual backbone in mobile ad-hoc networks needs to be re-established after a while, in order to adapt to the changing traffic conditions, network topology, as well as the battery levels of the mobile devices. The network topology may change slowly, whereas the battery levels of mobile devices and traffic conditions may change more frequently. Re-establishing the virtual backbone once a while could avoid draining the batteries of the members of the virtual backbone. Moreover, virtual backbone computation requires distributed algorithm to find the MWCDS, since there is no server or base-station in mobile ad-hoc network to perform such computation.

## H   APPENDIX: ML PIPELINES AND BASELINE HEURISTICS

### H.1   MWIS PIPELINE

We adopt the GCN-LGS pipeline in (Zhao et al., 2022a) as the actor network, which comprises a 3-layer GCNN followed by a distributed local greedy solver (LGS-MWIS) (Joo & Shroff, 2012). The actor GCNN is configured as follows: the dimensions of GCNN layers are $g_l = 32, l = 1, 2$, and $g_0 = g_3 = 1$, the hidden layers employ leaky ReLU activation, while the output layer ReLU activation. The GCNN outputs vector $\mathbf{c} \in \mathbb{R}^{|\mathcal{V}|}$. The local function in (5d) is instantiated as a multiplier, $f_{loc}(\mathbf{c}_i, \mathbf{z}_i) = \mathbf{c}_i \mathbf{z}_i$. The twin network is implemented by a 5-layer GCNN, in which the hidden layers have dimensions of $g_l = 64, l = 1, \dots, 4$ and leaky ReLU activation, and the output layer has a dimension of $g_5 = 2$ and Softmax activation. The training takes about an hour on a server with 16 GB memory, 8 CPUs and 1 GPU (Nvidia 1080 Ti).

The basic greedy heuristic first sorts the nodes by their weight in decreasing order, then iteratively adds a node to the solution and removes the node and its neighbors from the sorted list, until the list is empty.

The distributed heuristic LGS-MWIS (Joo & Shroff, 2012) iteratively builds a solution as follows: in an iteration, each node compares its weight with its neighbors; if a node has the maximal weight in the neighborhood, it marks itself as 1, and broadcasts a control message to its neighbors, who then mark themselves as -1; the unmarked nodes enter the next iteration. When all nodes are marked, the solution is the set of nodes marked as 1. LGS-MWIS has an average local communication complexity of $\mathcal{O}(\log |\mathcal{V}|)$ on general graphs. The centralized greedy algorithm and LGS-MWIS are detailed in (Zhao et al., 2022a, Algo 1 and Appendix Algo 1).

For MWIS problem in Section 4.1, the test set of 500 ER graphs and the corresponding optimal solutions are from `https://github.com/zhongyuanzhao/distgcn` (Zhao et al., 2022a).

### H.2   MWDS PIPELINE AND GREEDY-MWIDS

We augment Greedy-MWDS with a 5-layer actor GCNN and the same local function in the MWIS pipeline in Appendix H.1. The actor GCNN is configured as follows: the input and hidden layers have dimensions of $g_l = 32, l = 1, \dots, 4$ and leaky ReLU activations, the input and output dimensions are $g_0 = g_5 = 1$, and the output layer employs ReLU activation. The twin is also implemented by a 5-layer GCNN configured the same as the twin GCNN in the MWIS pipeline in Appendix H.1, except that $g_l = 32, l = 1, \dots, 4$. The GCNN is trained on randomly generated ER graphs of 100 to 300 nodes. The training takes about an hour on a server with 16 GB memory, 8 CPUs and 1 GPU (Nvidia 1080 Ti)

The greedy algorithm, Greedy-MWDS, iteratively builds a solution by adding to the solution the most cost-effective node from the vertices not yet in the solution, marking its neighbors as covered, until all nodes are either in the solution or covered. A good metric for the cost-effectiveness of node $v \in \mathcal{V}$ is (Jovanovic et al., 2010):

$$\omega(v) = \frac{c(v)}{1 + \sum_{u \in \mathcal{N}(v) \cap \mathcal{W}'} c(u)} , \mathcal{W}' = \{i | i \in \mathcal{V} \setminus (\mathcal{D}' \cup \mathcal{N}(\mathcal{D}'))\} , \tag{29}$$

where smaller $\omega(v)$ means better cost-effectiveness, $c(v)$ is the weight of node $v$, $\mathcal{D}'$ is the partial solution (initialized as $\varnothing$), $\mathcal{N}(\cdot)$ refers to the neighbors of a node or a vertex set, $\mathcal{W}'$ is the set of uncovered nodes.

An alternative greedy algorithm (Greedy-MWIDS) selects the most cost-effective node only from the uncovered vertices in each iteration, which builds an independent dominating set (IDS). Greedy-MWIDS can be implemented by the distributed LGS-MWIS (Joo & Shroff, 2012), and is used as part of the distributed heuristic for the MWCDS problem in Appendix H.4.

### H.3 NWST PIPELINE AND BASELINE HEURISTICS

The actor, twin, and local function are configured the same as in the MWDS pipeline in Appendix H.2. The input node feature for the actor and twin GCNNs is one-hot encoded indicator of whether a node is a terminal. The GCNNs are trained on randomly generated GRP graphs with 100 to 300 nodes. The training takes 5 hours on a server with 16 GB memory, 8 CPUs and 1 GPU (Nvidia 1080 Ti).

Shortest path heuristic (SPH) (Matsuyama, 1980) initializes the terminals as a set of subtrees; starting from an arbitrary subtree, it iteratively merges with its nearest subtree through the shortest path. The distance of a path is the total cost of nodes on it, where terminals have zero cost. The algorithm terminates when only one tree is left. The shortest path is found by Dijkstra's algorithm (Dijkstra et al., 1959). Kruskal's SPH (Bauer & Varma, 1996) (K-SPH) is a distributed variation of SPH, in which every subtree merges with its nearest subtree until only one tree is left. SPH has an approximation ratio of 2 (Matsuyama, 1980).

### H.4 MWCDS PIPELINE, BASELINE AND REFERENCE HEURISTICS

A low complexity heuristic for MWCDS problem (Sun et al., 2019) can be implemented in two steps: 1) find a MWDS, 2) connect the MWDS by solving a NWST problem where the terminals are the solution of step 1.

In our baseline heuristic, we choose Greedy-MWIDS in Appendix H.2 for step 1, and K-SPH-NWST in Appendix H.3 for step 2, and treat them as a single distributed greedy heuristic (Dist.Greedy).

Our reference algorithm is a centralized greedy heuristic, composed of Greedy-MWDS in Appendix H.2 for step 1 and SPH-NWST in Appendix H.3 for step 2.

The actor GCNN is configured as follows: the dimensions of input and hidden layers are $g_l = 32, l = 1, \ldots, 4$ and leaky ReLU activations, the input and output dimensions are $g_0 = 1, g_5 = 2$, and the output layer employs linear activation. The actor outputs $\mathbf{Z} \in \mathbb{R}^{|\mathcal{V}| \times 2}$, and the local function is a single neuron, $f_{loc}(\mathbf{c}_i, \mathbf{Z}_{i*}) = ReLU(\mathbf{Z}_{i,1}\mathbf{c}_i + \mathbf{Z}_{i,2})$. The twin network is a 5-layer GCNN configured identically as the twin in Appendices H.2 and H.3, except that its input dimension fits the output dimension of its own actor GCNN. The GCNN is trained on GRP graphs of 100 to 300 nodes. The training takes 8 hours on a server with 16 GB memory, 8 CPUs and 1 GPU (Nvidia 1080 Ti).

## I APPENDIX: CENTRALIZED RUNTIME IN SECONDS

We report the actual runtime of GCNN-enhanced COP solvers demonstrated in Section 4, measured on a laptop computer (Macbook Pro, 16GB memory, 2 GHz Quad-Core Intel Core i5, CPU only). It should be noted that these runtimes are based on our Python code, of which the implementation could be further optimized for running speed.

For the examples demonstrated by distributed COP solvers, the centralized runtime does not represent their distributed runtime. Therefore, we report both the total centralized runtime and the per-node centralized runtime for each distributed solver. The latter represents the estimated computational time in distributed execution. However, the major component of distributed runtime for distributed solvers is the communication time rather than the computational time, which could not be measured in our setting. In general, the GCNN no longer dominate the total runtime for these distributed COP solvers. For MWIS in Figure 9(a) and 9(b), the centralized runtime of distributed COP solvers are still linear to the graph size. For NWST (Figures 10(a) and 10(b)) and MWCDS (Figures 11(a) and 11(b)), the major component of the centralized runtime is Dijkstra's algorithm for the shortest path, which has a time complexity of $\mathcal{O}(|\mathcal{V}|^2)$ or $\mathcal{O}((|\mathcal{V}| + |\mathcal{E}|) \log |\mathcal{V}|)$.

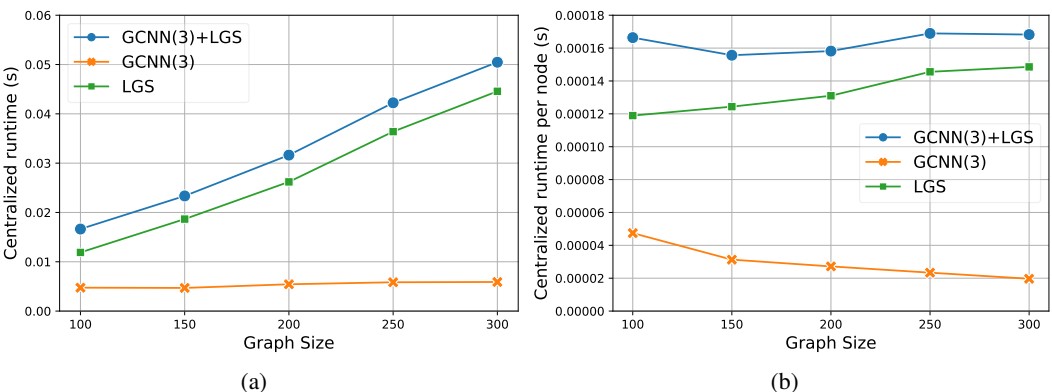

Figure 9: Average centralized runtime of GCNN-enhanced LGS-MWIS per instance, with a reusing factor $N = 1$. The total runtime is broken down to GCNN(3), as a one time overhead, and LGS, as the runtime for $N = \infty$. (a) Total centralized runtime by graph size, (b) Per node centralized runtime by graph size, representing the estimated computational time in distributed execution. Notice that the major component of distributed runtime is the communication time.

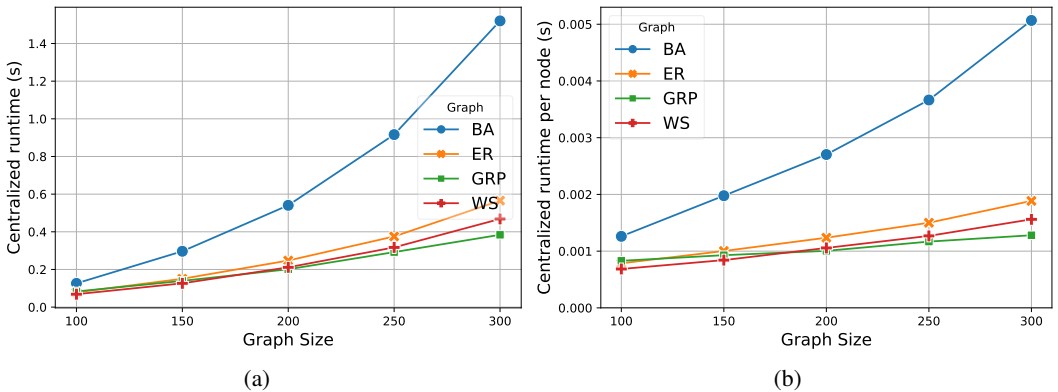

Figure 10: Average centralized runtime of GCNN-enhanced K-SPH-NWST solver per instance, with a reusing factor $N = 1$. (a) Total centralized runtime by graph size, (b) Per node centralized runtime by graph size, representing the estimated computational time in distributed execution. Notice that the major component of distributed runtime is the communication time.

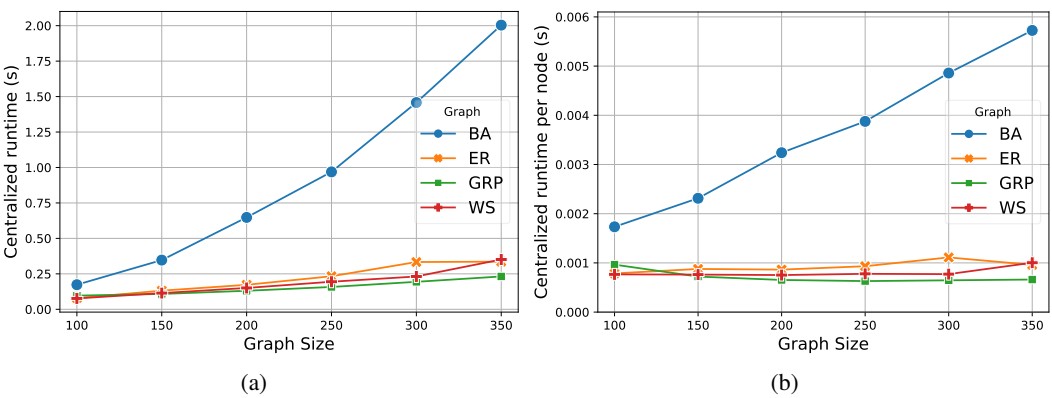

Figure 11: Average centralized runtime of GCNN-enhanced MWCDS per instance, with a reusing factor $N = 1$. (a) Total centralized runtime by graph size, (b) Per node centralized runtime by graph size, representing the estimated computational time in distributed execution. Notice that the major component of distributed runtime is the communication time.

