# OpenReview forum: "Graph-based Deterministic Policy Gradient for Repetitive Combinatorial Optimization Problems"
_ICLR.cc/2023/Conference — ICLR 2023 poster_

### Official Review · Reviewer_fA3E · 2022-10-19

**Confidence:** 4
**Correctness:** 4
**Technical Novelty And Significance:** 3
**Empirical Novelty And Significance:** 3
**Recommendation:** 8

**Clarity, Quality, Novelty And Reproducibility:**

This is a good paper with poor presentation.

- Certain variables are either undefined, or vaguely defined, or have inconsistent definitions. A non-exhaustive list:
    - What are the dimensions of $\mathbf{S}$ in the introduction? What do you mean by "data supported on $\mathcal{V}$"? Is $\mathbf{S}$ a matrix representation of the graph, node features, edge features...?
    - $f_{net}$, a key component of the optimization problem in (3), is not defined in Section 2.1, and neither is $f_{obj}$ in Section 2.2 (these are only vaguely defined in the introduction).
    - $\mathbf{o}$ is referred to as both a reward and a value function (although $\mathbf{r}$ is also used for the reward).
    - Both $\mathcal{G}(t)$ and $\mathbf{S}(t)$ are used to denote the state, and both $\mathbf{x}$ and $\mathbf{z}$ for the action. The difference between $\mathbf{Z}$ and $\mathbf{z}$ is also unclear.

Please correct inconsistent notations/definitions and either list them all before the technical sections, or make the technical sections self-contained.
- The parallels between the reformulations of the optimization problems in (3) and (5), and (4) and (10), are not clear. For example, from the text, it is not clear how "the objectives in (3a) and (5a) are equal due to the linearity of expectation". The variables that appear in these equations are not even the same. Please improve the description of these parallels, explaining how the different variables can be substituted.

Minor:

- Section 1 is too long and difficult to read. I suggest shortening the introduction, and using some of the space to describe the graph machine learning architectures used to parametrize GDPG-Twin.
- Use \citep for citations.
- What is the reusing factor in Figure 4?
- "Approximation ratio" (used to compare the proposed model with heuristic methods in the numerical experiments) is misleading, as it can be interpreted to mean that larger approximation ratio is always better. Consider referring to this quantity by another name.



**Details Of Ethics Concerns:**

N/A.

**Strength And Weaknesses:**

Strengths:

- The proposed learning framework is novel and broadly applicable.
- The numerical results are convincing:
    - On synthetic graphs, GDPG-Twin either improves upon or achieves faster computations than the conventional heuristics in four independent R-COPs, and optimizes the long-term objective in a graph MDP.
    - GDPG-Twin with random sampling around the current policy converges faster and has better sampling efficiency than the comparable method (ZOO).

Weaknesses:

- While the proposed framework is pretty general and is tested on four examples of COPs, many of the examples are very similar. It would have been interesting to see how the framework fares in different types of COPs, such as graph coloring (which also has applications in scheduling).
- The graph machine learning models used to parametrize GDPG-Twin are not described in detail. Moreover, the effect of the choice of parametrization is neither discussed nor illustrated with numerical experiments.
- Very little intuition is given as to the real-world applications of MWIS, MWIDS, NWST, and MWCDS and the differences among them. It is understandable that there is no real-world data for these problems, but a better description of their application to real-world problems would help ground the paper and strengthen its contribution.
- The paper is very hard to understand in its current form. See below.






**Summary Of The Paper:**

This paper introduces a novel formulation for repetitive combinatorial optimization problems (R-COPs) over graphs that is particularized to both independent R-COPs (i.e., problems where the graph data is independent at different time steps) and graph MDPs. It also introduces a novel actor-critic framework to solve these problems---GDPG-Twin---where the critic consists of a differentiable twin model that is trained to represents the states resulting from the application of a policy to a non-differentiable network process (typically based on a greedy heuristic).

**Summary Of The Review:**

The contributions of the paper are novel significant, but the presentation is poor. I recommend acceptance conditional on improving notation and presentation in the rebuttal.

---

> ### Author Response · Authors · 2022-11-12
> **Authors' reply (1)**
>
> We thank the reviewer for praising the novelty of this paper, as well as pointing out various issues in presentation. Since the main body of the paper already reaches the limit of 9 pages, it will take a bit longer for us to improve the presentation. We would like to first response to the reviewer's comments on the technical issues in this reply, while following up on the presentation issues with another reply and revised manuscript shortly.
>
> ## Weakness
> >  While the proposed framework is pretty general and is tested on four examples of COPs, many of the examples are very similar. It would have been interesting to see how the framework fares in different types of COPs, such as graph coloring (which also has applications in scheduling).
>
> The four examples of COPs are similar in the sense that they are all node-weighted problems. However, the underlying solvers of the four examples are different. In other words, we did not solve one type of COP by converting it to another type of COP in these four examples.
>
> Graph coloring problems are also very important, as you mentioned. It comes with at least 4 variations: Vertex coloring problem (VCP), equitable vertex coloring problem (ECP), Max-coloring (Max-Col) which can be seen as the weighted version of VCP, and Bin Packing Problem with Conflict (BPPC). See (Cornaz 2017) for their definitions. According to (Cornaz 2008, 2017), each of  these four variations can be converted in to maximum weighted independent set (MWIS) problem on an associated graph generated from the original graph in the graph coloring problem. For the weighted versions of graph coloring problems, such as Max-Col and BPPC, the node weights in the associated graph only depend on the node weights in the original graph, in other words, the topology of the associated graph would be the same if the topology of the original graph does not change.  Since we already demonstrated MWIS problem in our paper, our framework would be also applicable to repetitive Max-Col and BPPC (same underlying topology, changing weights). Notice that 'stable set' is just another name of 'independent set'.
>    - Denis Cornaz, Fabio Furini, Enrico Malaguti, Solving vertex coloring problems as maximum weight stable set problems, Discrete Applied Mathematics, Volume 217, Part 2, 2017, Pages 151-162, ISSN 0166-218X, [https://doi.org/10.1016/j.dam.2016.09.018](https://doi.org/10.1016/j.dam.2016.09.018).
>    - Denis Cornaz, Vincent Jost, "A one-to-one correspondence between colorings and stable sets," Operations Research Letters, Volume 36, Issue 6, 2008, Pages 673-676, ISSN 0167-6377, [https://doi.org/10.1016/j.orl.2008.08.002](https://doi.org/10.1016/j.orl.2008.08.002).
>
> >  The graph machine learning models used to parametrize GDPG-Twin are not described in detail. Moreover, the effect of the choice of parametrization is neither discussed nor illustrated with numerical experiments.
>
> Due to the limit of 9 pages of the main manuscript, we put the detailed description of the graph machine learning models of both actors and twin-based critics in appendixes, specifically, the equations of GCNN  in Appendix D, training datasets in Appendix E, and the hyperparameters of GCNN and the localized parameterization function in Appendix F.
>
> Indeed, there could be other types of graph neural networks that might work better than the GCNN used in the examples. Likewise, the hyperparameters or  parameterization function we have used could be improved. However, the main purpose of the experiments is to demonstrate the effectiveness of the proposed formulation and learning framework, rather than finding ML models and parameterization functions to achieve the best possible performance on those examples.
>
>
> > Very little intuition is given as to the real-world applications of MWIS, MWIDS, NWST, and MWCDS and the differences among them. It is understandable that there is no real-world data for these problems, but a better description of their application to real-world problems would help ground the paper and strengthen its contribution.
>
> The definition of each exemplary COP was explained in the corresponding paragraph of the result. Due to page limit, we will include more detailed explanations of the real-world applications of these four types of COPs in Appendix F.
>
> ## Minor
> >  Use \citep for citations.
>
> We will change accordingly.
>
> >  What is the reusing factor in Figure 4?
>
> The reusing factor N refers to the number of instances in repetitive COP that share the same underlying topology but have different sets of node weights, which can reuse the high-dimensional action generated by the actor network. For N=1, we run actor once per COP instance. For N=10, we run actor only once for every 10 instances. When the reusing factor N approaches to infinite, the computational and communication overheads of running the actor network for each COP instance become infinitely small.

---

> > ### Comment · Reviewer_fA3E · 2022-11-21
> > **Thank you**
> >
> > Thank you for clarifying that the underlying solvers of the four examples considered are different, and for explaining that other graph combinatorial problems (such as graph coloring) can be expressed in terms of the problems considered. I also appreciate the inclusion of details regarding the real-world applications of these problems, and the machine learning models that were used.
> >
> > I can see that the authors put a lot of work into explaining the notation and terminology and improving the problem description in both the rebuttal and the revision. All of my concerns have been addressed, so I will increase my score accordingly.

---

> ### Author Response · Authors · 2022-11-19
> **Authors' reply (2)**
>
> We thank the reviewer for the comments. In response to your comments on the poor presentation, we revised and uploaded the manuscript. In particular, the problem formulations of the two types of R-COPs are revised to truthfully reflect our goal and setting in this paper, meanwhile eliminate the ambiguity:
> 1. The problem formulations now include the practical restrictions that are the key settings of our paper but were not explicitly written out. In the solution, we explicitly state that the classical heuristic $h'$ (previous $h$) is given in advance to meet the practical restrictions. This would make the paper much easier to understand.
> 2. In the problem formulations and solutions, we replace all the vaguely defined $f_{net}$ by clearly defined policy network, such as $h$ for independent R-COPs and $\{h,\Psi\}$ for R-COPs in graph-based MDP.  In Introduction and Conclusion, most usages of $f_{net}$ are also replaced to more clearly terms, such as policy network. Now, $f_{net}$ only represents the entire network process. This improves the abuse of notation $f_{net}$ in the previous version, making the presentation more clear and accurate.
> 3. We modified the usage of several key terms (such as policy, action, etc.) and notations, which were vague, inconsistent or incorrect in the previous version. The modifications are listed below:
>
> ### New usage of notations:
>  - $\mathbf{S}$ is only referred as node features throughout the paper.
>  - $f_{net}$ only describes the entire network process, and is removed from the problem formulation and solutions.
>  - $f_{obj}$ still refers to an arbitrary known linear combination, to keep the generality of the proposed approach. It would only be clearly defined for a specific problem, such as in Section~4.2,  where$f_{obj}$ is defined as the average backlog across network and over time.
>  - $\mathbf{o}$ only refers to value vector
>  - $\mathbf{r}$ only refers to reward vector
>  - $\mathcal{G}$ and $\mathcal{G}(t)$ are referred as network state
>  - $\mathbf{z}(t)$ in  (10) is replaced by $\mathbf{c}(t)$  in this revision.
>
> ### New usage of terms
> - 'Policy' now refers to the function(s) that maps network state to decision. Specifically, in (3), the policy refers to the valid heuristic $h$. In (4), the policy refers to the combination of a valid heuristic $h$ and a cost function $\Psi$. In their reformulations (5) and (10), the 'policy' becomes the parameterized policy network. In (5), the policy network includes a given classical heuristic $h'$, local function $f_{loc}$ and actor network $\Psi$ (described after eqn (7)) in (5). In (10), the policy network includes a given classical heuristic $h'$ and the parameterized cost function $\Psi$ (which is also the actor network).
> - 'Action' is no longer used alone in the paper, but it implicitly refers to $\mathbf{x}$ or $\mathbf{x}(t)$, called "the vector of decisions" (or "decisions") . This make the graph-based Markov decision process easier to understand.
> - 'Intermediate action' is now used to refer $\mathbf{Z}$ in (5) and $\mathbf{c}(t)$ in (10), since they are the outputs of the actor networks, meanwhile respectively belong to the overall parameterized policy networks.
> - 'Policy parameters' is now used for $\mathbf{Z}$ in (5) since it parameterizes the local function $f_{loc}$ and subsequently the given classical heuristic $h'$.
> - 'Actor' or 'actor network' refer to $\Psi$ in (5) and (10) that generates 'intermediate action'. 'Actor network' is the trainable part of the parameterized policy network.
> - In Section 3.1 (solution for independent R-COPs), we use 'Policy gradient' for $\nabla_{\mathbf{Z}}\boldsymbol{1}^{\top}\mathbf{o}$, and use 'actor gradient' for $\nabla_{\mathbf{\Theta}_p}\boldsymbol{1}^{\top}\mathbf{o}$, since the actor could be absent, i.e., for R-COPs on fixed topology.
> - In Section 3.2 (solution for R-COPs in graph-based MDP), 'policy gradient' and 'actor gradient' refer to the same thing $\nabla_{\mathbf{\Theta}_p}$, since the actor network is the only trainable part of the policy network.
>
> We believe that the usage of terms and notations in this revision is more consistent and self-contained.
>
> ## The parallels between the reformulations of the optimization problems in (3) and (5), and (4) and (10)
>
> In (5), we now explain the equivalence between the objectives in (3a) and (5a) by adding mathematical proof
>
> $E(1^\top x)=E(\sum_{i=1}^{|c|}c_i x_i)$ $= \sum_{i=1}^{|c|} E(c_i x_i) = 1^{\top}o$
>
> to the original sentence.
>
> In the revised problem formulation (4), we describe the roles of $h$ in (4d) and $\Psi$ in (4e), which are now consistent with their parameterized version in (10d) and (10e), respectively. Now the combination $\{h,\Psi\}$ in (4) and (10) are policy and parameterized policy network, respectively. This better describes the parallels between (4) and (10).
>
> (to be continued)

---

> ### Author Response · Authors · 2022-11-19
> **Authors' reply (3)**
>
> ## Detailed description of the graph machine learning models used in GDPG-Twin
>
> We added more detailed description of the hyperparameters of the GCNNs for both actor and twin networks in each of the four ML pipelines in Appendix G (previously, F).
>
> We also revised the sentences in the main manuscript to make it more clear for the readers in finding the related details of the GDPG-Twin pipelines in three Appendices D, E, G, as follows:
> > The details of the four ML pipelines are in Appendix D for the equations of GCNN, Appendix E for the detailed training configurations, and Appendix G for the hyperparameters of GCNNs and brief descriptions of the chosen baseline heuristics.
>
> ## Description of the applications of four exemplary COPs in real-world problems
>
> In the newly added Appendix F, we include two pages of description for the applications of the four exemplary COPs in real-world problems, along with citations. We also updated the citations referring to their real-world applications in the first paragraph of Section 4.1 in the main manuscript.
>
> ## Section 1 is too long and difficult to read.
>
> We revised the Introduction (Section 1), to make it easier to follow, and a little bit shorter. The first paragraph is revised to better describe the setting and goal of our work. We also modified the corresponding description of our contribution, and how our approach fill the gaps in the existing work (The last paragraphs in Sections 1.1 and 1.2, respectively) according to the revised problem formulation.
>
> We believe these modifications can improve the readability of Section 1.
>
> ## The use of "Approximation ratio", "as it can be interpreted to mean that larger approximation ratio is always better"
>
> In the revised manuscript, we place  "(larger is better)" or "(Smaller is better)" right after 'Approximation ratio' in the title of corresponding figures, to avoid misinterpretation.
>
> In the literature of COPs that minimizes the objective function, such as Steiner tree problem, MWDS, and MWCDS, 'approximation ratio' has been used to refer 'smaller is better'. Conventionally, 'approximation ratio' does not always mean the larger is always better.
>
> For example, in (Robins, 2005), the authors claim in the Introduction that "the best Steiner approximation ratio ... was gradually improved from 2 in (Matsuyama, 1980) to 1.59 in a series of papers", and the authors further improved it to 1.55.
> Another example in (Erlebach, 2010), the authors claim they improve the approximation ratio of MWDS heuristic from (5+$\epsilon$) to (4+$\epsilon$).
> - Robins, G., & Zelikovsky, A. (2005). Tighter bounds for graph Steiner tree approximation. _SIAM Journal on Discrete Mathematics_, _19_(1), 122-134.
> - A Matsuyama. An approximate solution for the steiner problem in graphs.  Math. Japonica, 24:573–577, 1980.
> - Erlebach, T., Mihalák, M. (2010). A (4 + _ε_)-Approximation for the Minimum-Weight Dominating Set Problem in Unit Disk Graphs. In: Bampis, E., Jansen, K. (eds) Approximation and Online Algorithms. WAOA 2009. Lecture Notes in Computer Science, vol 5893. Springer, Berlin, Heidelberg. https://doi.org/10.1007/978-3-642-12450-1_13
>
>
> The original meaning of 'approximation ratio' is just the ratio between a quantity of interest and a reference quantity. The convention in the literature to use 'approximation ratio' to compare the heuristic with the optimal solver in the literature. To avoid such misunderstanding, we explicitly use 'approximation ratio w.r.t. greedy' and 'approx. ratio w.r.t. optimal' in the text as well as the figures.
>
> We think these measures could void readers misinterpreting our results. If the reviewer could suggest a better term, we would consider to use it.
>
> We believe that we have respond to all the comments of the reviewer. Again, we thank the reviewer for helping us improve the paper. We hope that these responses and the revised manuscript could address most of the concerns of the reviewer.

---

### Official Review · Reviewer_XHWP · 2022-10-25

**Confidence:** 2
**Correctness:** 3
**Technical Novelty And Significance:** 3
**Empirical Novelty And Significance:** 3
**Recommendation:** 8

**Clarity, Quality, Novelty And Reproducibility:**

The clarity, quality, and novelty of the paper are good. The author shares the code for reproducing.

**Strength And Weaknesses:**

Strength:
The proposed method is novel and gives clear improvements compared to existing methods. The introduction and literature review gives a good context of the problem. The experiment results are strong.

Weakness:
Could improve the citation style.

**Summary Of The Paper:**

This paper proposes a novel GNN and RL-based approach to R-COPs with hard constraints and without supervised labels. The proposed approach has several advantages compared to existing methods and show clear improvements emperically.

**Summary Of The Review:**

The paper is well-written and the proposed method is novel and has clear improvements.

---

> ### Author Response · Authors · 2022-11-19
> **Authors' reply**
>
> We thank the reviewer for spending time on our work and the high recommendation.
> The citation style has been fixed in the revised manuscript.

---

### Official Review · Reviewer_gr9w · 2022-10-26

**Confidence:** 4
**Correctness:** 3
**Technical Novelty And Significance:** 3
**Empirical Novelty And Significance:** 3
**Recommendation:** 6

**Clarity, Quality, Novelty And Reproducibility:**

**Clarity & Quality**
The paper is not clear to me, mainly because of the mathematical issues with the very initial formulations which did not help to understand what was exactly the problem being addressed.

**Novelty**
I did not understand the paper enough to be able to judge precisely.

**Reproducibility**
The source code is provided.

**Remarks/questions**
* Sec 1.1. “network state of a time slot. ” —> at a time solt?
* The reference for the Branch and Bound algorithm should be fixed. The paper references recent improvements (2019-2021 ML papers) while B&B was invented in the 60s in the OR community.
* Sec 2.1: “the smoothened objective function in (3a) makes it easier to approximate f_net^*, as illustrated in Section 3.” Can the paper be more specific about how it is illustrated?
* Sec 3.1.1. In Actor Critic algorithms, we always sample from the policy to generate the next action (for exploration). What is the argument for introducing a perturbation of Z instead of sampling Z from the current policy?


**Strength And Weaknesses:**

**Strengths**
1. The paper cites and explains a lot of related works
1. The approach is tested on 4 problems

**Weaknesses**
1. Important issues with the formalization of the problem or the notations, that make understanding the primal goal/setting of the paper difficult:
    * The initial problem Eq (1) optimize f_{obj}(o) where o = f_net(G) but G is the network state that looks like the input. So I don’t see what is the decision variable here
    * Sec 2: “We define a function space F as a set of functions that will output solutions satisfying all the constraints of the COP in (2).” What is the input of these functions?
1. Some mathematical statements lack rigor or are wrong in my opinion :
    * Sec 2.1: We relax the objective function in (2a) to its expectation over Ωc. How is this a relaxation (in the mathematical optimization sense)? Problem (3a-b) is another optimization problem, where the decision variable is F_net
    * Sec 2.1 “optimal solution for (3), x^∗ = f_net^*(V, E, c), is still an exact solver for (2) for every given c”.  f_net^* is by definition the optimal solution when minimizing the expected objective over c. Therefore for a given c, there is no guarantee that it is optimal.
    * Sec 3.1. The fct h in eq (5) is presented as a valid heuristic, but it looks like a constraint in (5c). Is it an input ? (in which case h \in F should not appear as a constraint);  Or a variable ? (in this case how is it linked to the decision variable Z?)
1. The paper is very dense and was not easy to read for me — although I have a strong background in CO and ML but admittedly not so much in network problems specifically.
    * E,g. Sec 1.1 could use some more structure like paragraphs
    * I don’t understand: Sec 1.2: “Imitation learning …. However, it requires a good guiding algorithm, which set the upper limit of the system and could be costly”
    * Even after reading the paper carefully, the first sentence of the conclusion  is still hard to get "...address repetitive combinatorial optimization problems by parameterizing the input node weights of a non-differentiable, fast and/or distributed heuristic, fnet(·), through a continuous-valued high-dimensional action generated by an actor GNN"



**Summary Of The Paper:**

The paper proposes an RL approach to address two families of CO problems: (i) independent repetitive COPs and (ii) COPs embedded in graph-based MDPs. It claims to learn reusable node and edge information to improve existing heuristics to these problem. The proposed approach is illustrated on graph CO problems.


**Summary Of The Review:**

I would vote for reject because the paper is not clear and lacks mathematical correctness.


******* **After Rebuttal** ******

I thank the authors for their replies. After reading the revised manuscript and the extensive rebuttal, I think the paper is more clear and many of my questions were answered. I still have a few minor doubts/remarks:

* The first statement of the abstract: “We propose an actor-critic framework for graph-based machine learning pipelines with non-differentiable blocks, and apply it to repetitive combinatorial optimization problems (COPs) under hard constraints” but the paper really focuses on repetitive COPs and the framework is only defined for repetitive COPs.
* In Eq (4), should the sets of vertices $\mathcal{V}(t)$ and edges $\mathcal{E}(t)$ depend on $t$? They don’t seem to be updated for different $t$s. Only $S(t+1)$ is defined in Eq (4f).
* In (4b), since the expectation is taken with respect to the heuristic $h$ and cost $\psi$, are these functions stochastic?
* This expectation in (4b) should also depend on $f_r$ and $f_s$ which are defined as stochastic.
* In (3a), $h$ does not appear in the expectation — is it deterministic there?
* The paper says that the local function in (5d) “…can be chosen as, e.g., a multiplier, a single neuron, or even a small neural network,..”  If it is a neural model then it is not clear how it can be learned within the framework. I see in the Appendix that in the experiments the local function is “a multiplier”: $f_{loc}(c_i, z_i) = c_i z_i$ — therefore no learning needed. I think the paper should explain how $f_{loc} $ can be trained or remove the statement that it can be a neuron/neural network.

Details:
* In Eq (7) $\bar{c}$ could be used for readability.
* Sec 2.2: “$0 \leq \gamma \leq 1$”: the discount factor should be strictly positive
* Sec 3.1 “$0 \leq \alpha_p \leq 1$": similarly the learning rates $\alpha_p$ and $\alpha_c$ should be strictly positive, and I’m not sure about why the paper uses 1 as an upper bound. Of course in practice the learning rate is much smaller than 1 but I don’t think it **has** to be smaller than 1.

Overall I think the paper presents an interesting approach to improve the performance of heuristics for repetitive COPs over graphs. The proposed framework is successfully demonstrated on 5 problems. Therefore I am happy to increase my score and confidence level.

---

> ### Author Response · Authors · 2022-11-12
> **Authors' Rebuttal (1)**
>
> We are sorry that the mathematical issues with the very initial formulations led to an incomplete understanding of the problem being addressed and did not allow you to properly evaluate our paper. We would do our best to clarify it through itemized responses to your questions and comments. We will update the manuscript accordingly and upload it soon.
>
> > 1. Important issues with the formalization of the problem or the notations, ... understanding the primal goal/setting of the paper difficult:
> > ○	The initial problem Eq (1) optimize f_{obj}(o) where o = f_net(G) but G is the network state that looks like the input. So I don’t see what is the decision variable here
>
> Indeed, $\mathcal{G}$ is the input of $f_{net}(\cdot)$. To optimize $f_{obj}(\mathbf{o})$, the decision variable is the network process $f_{net}(\cdot)$, often just part(s) of it in practice, e.g., we can not change the constraints of a combinatorial problem but can improve the solver.
>
> > ○	Sec 2: “We define a function space F as a set of functions that will output solutions satisfying all the constraints of the COP in (2).” What is the input of these functions?
>
> The input of these functions is the network state $(\mathcal{V}, \mathcal{E}, \mathbf{c})$.
>
> > 2. Some mathematical statements lack rigor or are wrong in my opinion :
> > ○	Sec 2.1: We relax the objective function in (2a) to its expectation over Ωc. How is this a relaxation (in the mathematical optimization sense)? Problem (3a-b) is another optimization problem, where the decision variable is F_net
>
> Strictly speaking, (3) is a re-formulation of (2) under the repetitive COP settings, rather than a relaxation. In (2), we seek to find the optimal solution for one COP instance. In (3), we seek to find a function $f\in\mathcal{F}$ that can find the optimal solutions for all the instances of a COP problem. Therefore, although the decision variables in these two formulations are different, optimizing (3) is equal to optimizing (2). This equivalence is explained in the next response.
>
> > ○	Sec 2.1 “optimal solution for (3), x^∗ = f_net^*(V, E, c), is still an exact solver for (2) for every given c”. f_net^* is by definition the optimal solution when minimizing the expected objective over c. Therefore for a given c, there is no guarantee that it is optimal.
>
> By definition, the objective (wrote in multi-lines for correct display)
>
> $\mathbb{E}_{c\sim\Omega^{c}} (c^\top x)$
>
> =
>
> $\underset{N\rightarrow\infty}{\lim}\frac{1}{N}\sum_{n=1}^{N}{c}(n)^\top x(n)$, for all $c(n)\sim\Omega^c$.
>
> Let’s say we have $x^*(n)=f_{net}^*(V,E,c(n))$ and $x^1(n)=f_{net}^1(V,E,c(n))$, where $x^*(n)$ is the optimal solution for every instance, whereas $x^1(n)$ is the optimal solution for every instance except the one with a given $c(i)$, that is $c(i)^\top x^*(i)<c(i)^\top x^1(i)$ in a minimization problem.
>
> Then we have
>
> $\mathbb{E}_{c\sim\Omega^{c}}(c^\top  x^*)$
>
> <
>
> $\mathbb{E}_{c\sim\Omega^{c}}(c^\top  x^1)$
>
> Therefore, $x$ needs to be always the optimal solution for any input $c\sim\Omega^{c}$, in order to optimize $ \mathbb{E}_{c\sim\Omega^{c}}(c^Tx) $.
>
> > ○	Sec 3.1. The fct h in eq (5) is presented as a valid heuristic, but it looks like a constraint in (5c). Is it an input ? (in which case h \in F should not appear as a constraint); Or a variable ? (in this case how is it linked to the decision variable Z?)
>
> The valid heuristic $h$ is given a priori, usually based on practical constraints such as runtime and/or distributed implementation, so it is considered as an input to (5). The heuristic $h$ is not a variable to be optimized.
>
> > 3. The paper is very dense and was not easy to read for me ...
> > ○	E,g. Sec 1.1 could use some more structure like paragraphs
>
> Could you be a bit more specific about this point?
>
> > ○	I don’t understand: Sec 1.2: “Imitation learning …. However, it requires a good guiding algorithm, which set the upper limit of the system and could be costly”
>
> Imitation learning is a supervised learning method that generates training labels on-the-fly by solving each instance of the problem with a given guiding algorithm. Therefore, the guiding algorithm sets the upper limit of the ML pipeline. For the ML pipeline to perform well, it requires a guiding algorithm with good performance, which often comes with a high computational cost.
>
> > ○	Even after reading the paper carefully, the first sentence of the conclusion is still hard to get "...address repetitive combinatorial optimization problems by parameterizing the input node weights of a non-differentiable, fast and/or distributed heuristic, fnet(·), through a continuous-valued high-dimensional action generated by an actor GNN"
>
> We would like to revise the expression as:
> “We propose to address repetitive combinatorial optimization problems with a fast and/or distributed heuristic, $f_{net}(\cdot)$, of which the input node weights are parameterized by a continuous-valued high-dimensional action generated by an actor GNN.”
>
> (To be continued)

---

> ### Author Response · Authors · 2022-11-12
> **Authors' rebuttal (2)**
>
> (Continued from rebuttal (1))
>
> > ... make understanding the primal goal/setting of the paper difficult...
>
> The primal goal/setting of this paper is to optimize a network-wide system objective, which is the outcome of some network process. The decision variable of the optimization is (part of) the network process. Specifically, the network process involves repetitively solving instances of certain combinatorial optimization problem on a graph (network). These repetitive COP instances share the same underlying graph topology, but differ in node weights. Moreover, there are practical constraints on how to solve these COP instances, such as limited per-instance runtime and/or distributed implementation of the COP solver. Additionally, the network process could also involve graph-based MDP.
>
> Our goal is to provide a general framework, including problem formulation and learning approach, that can:
> 1. Improve the average quality of solution for each repetitive COP instance generated by a fast and/or distributed COP solver (which could be the only option to meet the practical constraint), with minimal increases in per-instance runtime, meanwhile, if applicable, preserving the distributed implementation of the solver. (In our experiments, 3 out of 4 examples are based on fully distributed COP solver.)
> 2. Improve the long-term objective for a network process that involves graph-based MDP, meanwhile preserving the distributed implementation of the solver if applicable.
>
> ## Remarks/questions
>
> > Sec 1.1. “network state of a time slot. ” —> at a time solt?
>
> Thanks for pointing out this, we now change it to ‘at a time slot’
>
> >The reference for the Branch and Bound algorithm should be fixed. The paper references recent improvements (2019-2021 ML papers) while B&B was invented in the 60s in the OR community.
>
> Thanks for pointing out this, we now added the following reference of B&B: “Ailsa, H., Land., Alison, G., Doig. An Automatic Method for Solving Discrete Programming Problems. Econometrica, (1960).;28(3):105-132. doi: 10.1007/978-3-540-68279-0_5”
>
> >Sec 2.1: “the smoothened objective function in (3a) makes it easier to approximate f_net^*, as illustrated in Section 3.” Can the paper be more specific about how it is illustrated?
>
> The objective function in (3a), smoothened by the expectation of the weighted sum of decisions, allows us to break down the objective function into the sum of expectations in (5a) and (5b). This allows us to predict the element-wise expected outcomes $\mathbf{o}$ through the twin-based critic network. The twin-based critic network then generates gradients in backpropagation to optimize the actor network, which generates high-dimensional action to improve the decisions.
>
> > Sec 3.1.1. In Actor Critic algorithms, we always sample from the policy to generate the next action (for exploration). What is the argument for introducing a perturbation of Z instead of sampling Z from the current policy?
>
> The arguments of introducing a perturbation of Z include two points, which have been discussed in paragraph 1 of Section 3.1.1:
> 1) Some repetitive COPs could be defined on a fixed network topology, such as a wired telecom network, microwave backbone network, road network, etc., where $Z$ would be a constant vector/matrix. In such cases, we won’t need an actor to generate a high-dimensional action $Z$ because an actor will always generate the same $Z$ given the same input topology. In such fixed-topology scenarios, we need to introduce perturbation to allow the twin-based critic to learn the loss landscape around the current $Z$, and generate gradient in the backpropagation to optimize $Z$.
> 2) Introducing a perturbation of $Z$ could increase the sampling efficiency of the twin-based critic in learning the loss landscape around current $Z$ (or policy).
>
> We thank the reviewer for spending time on this work. We will soon upload a revised manuscript that incorporate updates that would help clarify the issues mentioned by the reviewer.
>
> Meanwhile, we would be happy to address any followup comments from the reviewer if these responses could help with the assessment of the technical correctness and contributions of this paper.

---

> ### Author Response · Authors · 2022-11-19
> **Authors' reply (3) the problem formulation, notations and terms are improved**
>
> We thank the reviewer for pointing out the various problems in the problem formulation and mathematical description. In response to your comments, we significantly improved the Introduction, problem formulation, usage of terms and notations in the revised manuscript, which has been uploaded.
>
> To better describe our primal goal and setting in this paper, we revised the first paragraph of the paper by stating that 'We aim to optimize the system-level objective ...  by improving parts of the network process $f_{net}$.'
>
> Moreover, in the revised problem formulations (Section 2), we introduce space $\mathcal{P}$ of functions under practical restriction to better reflect our goal and setting, and we also clearly stated in solution (Section 3) that a baseline heuristic $h'$ is given in advance to meet the practical restrictions.
>
> The modification in this revision is summarized as follows:
> 1. The problem formulations now include the practical restrictions that are the key settings of our paper but were not explicitly written out. In the solution, we explicitly state that the classical heuristic $h'$ (previous $h$) is given in advance to meet the practical restrictions. This would make the paper much easier to understand.
> 2. In the problem formulations and solutions, we replace all the vaguely defined $f_{net}$ by clearly defined policy network, such as $h$ for independent R-COPs and $\{h,\Psi\}$ for R-COPs in graph-based MDP.  In Introduction and Conclusion, most usages of $f_{net}$ are also replaced to more clearly terms, such as policy network. Now, $f_{net}$ only represents the entire network process. This improves the abuse of notation $f_{net}$ in the previous version, making the presentation more clear and accurate.
> 3. We modified the usage of several key terms (such as policy, action, etc.) and notations, which were vague, inconsistent or incorrect in the previous version. The modifications are listed below:
>
> ### New usage of notations:
>  - $\mathbf{S}$ is only referred as node features throughout the paper.
>  - $f_{net}$ only describes the entire network process, and is removed from the problem formulation and solutions.
>  - $f_{obj}$ still refers to an arbitrary known linear combination, to keep the generality of the proposed approach. It would only be clearly defined for a specific problem, such as in Section~4.2,  where$f_{obj}$ is defined as the average backlog across network and over time.
>  - $\mathbf{o}$ only refers to value vector
>  - $\mathbf{r}$ only refers to reward vector
>  - $\mathcal{G}$ and $\mathcal{G}(t)$ are referred as network state
>  - $\mathbf{z}(t)$ in  (10) is replaced by $\mathbf{c}(t)$  in this revision.
>
> ### New usage of terms
> - 'Policy' now refers to the function(s) that maps network state to decision. Specifically, in (3), the policy refers to the valid heuristic $h$. In (4), the policy refers to the combination of a valid heuristic $h$ and a cost function $\Psi$. In their reformulations (5) and (10), the 'policy' becomes the parameterized policy network. In (5), the policy network includes a given classical heuristic $h'$, local function $f_{loc}$ and actor network $\Psi$ (described after eqn (7)) in (5). In (10), the policy network includes a given classical heuristic $h'$ and the parameterized cost function $\Psi$ (which is also the actor network).
> - 'Action' is no longer used alone in the paper, but it implicitly refers to $\mathbf{x}$ or $\mathbf{x}(t)$, called "the vector of decisions" (or "decisions") . This make the graph-based Markov decision process easier to understand.
> - 'Intermediate action' is now used to refer $\mathbf{Z}$ in (5) and $\mathbf{c}(t)$ in (10), since they are the outputs of the actor networks, meanwhile respectively belong to the overall parameterized policy networks.
> - 'Policy parameters' is now used for $\mathbf{Z}$ in (5) since it parameterizes the local function $f_{loc}$ and subsequently the given classical heuristic $h'$.
> - 'Actor' or 'actor network' refer to $\Psi$ in (5) and (10) that generates 'intermediate action'. 'Actor network' is the trainable part of the parameterized policy network.
> - In Section 3.1 (solution for independent R-COPs), we use 'Policy gradient' for $\nabla_{\mathbf{Z}}\boldsymbol{1}^{\top}\mathbf{o}$, and use 'actor gradient' for $\nabla_{\mathbf{\Theta}_p}\boldsymbol{1}^{\top}\mathbf{o}$, since the actor could be absent, i.e., for R-COPs on fixed topology.
> - In Section 3.2 (solution for R-COPs in graph-based MDP), 'policy gradient' and 'actor gradient' refer to the same thing $\nabla_{\mathbf{\Theta}_p}$, since the actor network is the only trainable part of the policy network.
>
> We believe that these modifications can improve the inaccurate or problematic mathematical statements in the previous version.
>
> In the next reply, we would like to provide itemized explanations on how your comments are addressed in this revision.
> (to be continued)

---

> ### Author Response · Authors · 2022-11-19
> **Authors' reply (4) how the reviewer's comments are addressed in the revised manuscript**
>
> ## 1.1 The initial problem statement, unclear decision variable
> We modified the initial problem statement to make it clear that the decision variable is part of the network process $f_{net}$
> > We aim to optimize the system-level objective $f_{obj}(\mathbf{o})$, where $f_{obj}:\reals^{|\mathbf{o}|}\rightarrow \reals$ is a known linear combination, by improving parts of the network process $f_{net}(\cdot)$.
>
> ## 1.2 Input of the functions in space $\mathcal{F}$
> We restricted the use of $f_{net}$, which no longer appears in Sections 2 and 3. The function space $\mathcal{F}$ now only refers to valid heuristics to COP in (2), with its input given as follows:
>
> > Furthermore, we define the function space $\mathcal{F}$ of valid heuristics, meaning that  $\mathbf{x}=f(V,E,\mathbf{c})$ satisfies the constraints in (2) for all $f\in\mathcal{F}$.
>
> We now only refer heuristics $h\in\mathcal{F}$ and $h'\in\mathcal{F}$ through out the paper.
>
> ## 2 Mathematical statements lack rigor or are wrong
>
> ### 2.1 Problem formulation from (2) to (3)
> In the revised manuscript, we no longer call it relaxation in problem formulation in (3).
> ### 2.2 "Optimal solution for (3), ..., is still an exact solver for (2)"
> Since we have revised the formulation, this statement is no longer true and has been removed.
> ### 2.3 'h' is an input or variable in (5)
> In this revision, we use $h'$ to represent a given baseline heuristic in (5), and we also explicitly state that $h'$ is given in advance.
> In the revised (3) and (4), however, $h$ is a decision variable, which is clearly stated in the equations.
>
> ## 3 The paper is difficult to read
> ### 3.1 use structure like paragraphs for Sec 1.1
> We revised the Sec 1.1 and introduce bold text "Centralized solvers" and "Distributed solvers" to add more structure to the first and second paragraphs. In addition, we also improve the description to make it more brief.
>
> ### 3.2 statement about "imitation learning"
> We removed most of those problematic description, and use the following statement in the discussion of "imitation learning"
>
> > In some scenarios, such non-differentiable pipeline can be circumvented by using soft constraints (Kendall, 1975), which allows a fully differentiable policy network that can be trained by policy gradient-based primal-dual optimization (Eisen et al., 2019) or imitation learning (Ross et al., 2011).
>
> ### 3.3 The first sentence of conclusion
> After revising the problem formulation, usage of terms and notations, the first sentence of the conclusion now becomes:
>
> > We address repetitive combinatorial optimization problems under practical restrictions in runtime and/or distributed execution, by introducing a non-differentiable policy network based on a hand-picked, fast and/or distributed heuristic, which is parameterized by a continuous-valued high-dimensional intermediate action from an actor GNN.
>
> ## Remarks/Questions
> ### "network state at a time slot"
> This is fixed in this revision.
>
> ### Reference for branch-and-bound
> In this revision, the reference is fixed according to the reviewer's comment.
>
> ### explanation of the "smoothened objective function in (3a)"
> We revise the formulation in (3), this statement has been removed.
>
> ### Argument for introducing a perturbation of Z
> Please notice that the perturbation of Z is only used during training. In execution, the action is only based on current Z.
> We now better explain the reason of introducing of a perturbation of Z in the beginning of Sec 3.1.1
>
> > For applications based on static topologies, i.e., $(V, E,\bar{\mathbf{c}})$ are constant, we no longer need an actor to generate $\mathbf{Z}$. In this case, the twin is likely to be overfitted if we only feed it with a static $\mathbf{Z}$ during training. To address this problem, ...
>
>
> We believe that we have respond to all the comments of the reviewer. Again, we thank the reviewer for helping us improve the paper. We hope that the revised manuscript could address the reviewer's concern in presentation and mathematical rigor and correctness, and better assess the contribution of our work.

---

### Decision · Program_Chairs · 2023-01-20

**Decision:**

Accept: poster

**Justification For Why Not Higher Score:**

Neither the method nor the overall importance of the problem was articulated clearly enough to, in its current form, be interpretable and of interest to a substantial subset of the ICLR audience.

**Justification For Why Not Lower Score:**

Support from two knowledgable reviewers

**Metareview: Summary, Strengths And Weaknesses:**

This paper formulates the problem of solving repetitive combinatorial optimization problems as a graph-based MDP and then learning a policy to assign values over time based on the evolution of the problem.  It accomplishes this by learning a twin of the non-differentiable part of the model, and then performing policy optimization with respect to the system.

The method is demonstrated to be effective four classical combinatorial optimization problems, tested on a variety of different classes of network topologies.  It seems always to be helpful, marginally so in some topologies and substantially so on others.

The reviewers and AC found the paper unclear or difficult in many aspects.  The authors answered in detail and allayed the concerns of one of the reviewers, but some substantial concerns remain.

Overall, the paper opens up an interesting and important set of problems but the formulation and explanation remain very difficult and it is unlikely that this paper will have impact.   Here are some additional suggestions for improvement:
- Make clear how the constraints mentioned in (2c) are related to the graph structure
- Are the constraints all only of arity 2?   If not, wouldn't a hypergraph network be a better (more general) model?
- One novel idea here (which is cool!) gets buried, but deserves emphasis:  (as I understand it) in the MDP formulation, the effective "controls" are actually cost vectors for COPs to be solved at each time step (using a parameterized "heuristic" method h).
- It would help a lot to frame an example at the beginning:  in particular, it's a bit difficult to disentangle the meanings of s, x, o, r, and c.  My understanding is (in the independent case):  s is an observable state of the system, x is a "low-level" control, o is a vector of outcomes (of applying control x in state s), c is a linear weighting of the o values to obtain a scalar objective.
- But then, in the MDP case, we have also f_obj and r.   I understand now that we are going to describe a policy by specifying a cost function c and a heuristic function.  I also understand, I guess, that we are trying, ultimately, to optimize f_obj (in 4a).  But:
  o   I don't understand how that relates to f_r
  o   I'm not sure I see why it makes sense to apply f_obj to the *expected* (averaged over time?) o values?
- Section 3.1 is hard to decode.  In the end, is the idea that the Z values are parameters of h (indirectly, via f_loc)?   And then this is a straight-up optimization problem (not sequential)?
- In the experimental section, begin by listing the claims you want to make about your method, and then provide clear and specific data and arguments supporting those claims.  In the discussion of the number of rounds for convergence---is this about training time or performance time?   It's not at all clear what these "rounds" are.

In summary, there is interesting work here, but to have real impact it will be important to continue to work to improve the exposition of both the formulation and the empirical results.

**Note From Pc:**

if the above contains the word "oral" or "spotlight" please see: "oral" presentation means -> notable-top-5% and "spotlight" means -> notable-top-25%. As stated in our emails, we are disassociating presentation type from AC recommendations

---

> ### Author Response · Authors · 2023-03-01
> **Updates in the camera ready version**
>
> We thank the program chairs for the final decision and these important suggestions for improving the paper. We addressed each of these suggestions in the camera ready version.
>
> > Make clear how the constraints mentioned in (2c) are related to the graph structure
>
> In the discussion after (2), we added description that the constraints in (2c) are typically defined on graphs and gave an example to explain it.
>
> > Are the constraints all only of arity 2? If not, wouldn't a hypergraph network be a better (more general) model?
>
> In the discussion after (2), we added descriptions that the arity of discrete constraints in (2b) does not have to be 2, and that the constraints in (2c) could be defined on higher-order networks. Such a generalized form of (2) can still be captured by the defined function space of valid heuristics, $\mathcal{F}$, in the problem formulations and the solutions.
>
> > One novel idea here (which is cool!) gets buried, but deserves emphasis: (as I understand it) in the MDP formulation, the effective "controls" are actually cost vectors for COPs to be solved at each time step (using a parameterized "heuristic" method h).
>
> We highlighted this idea in the first contribution, as well as the corresponding discussion in other parts of introduction.
>
> > It would help a lot to frame an example at the beginning: in particular, it's a bit difficult to disentangle the meanings of s, x, o, r, and c. My understanding is (in the independent case): s is an observable state of the system, x is a "low-level" control, o is a vector of outcomes (of applying control x in state s), c is a linear weighting of the o values to obtain a scalar objective.
>
> We added a new appendix (Appendix A in the camera ready version), providing two exemplary formulations of wireless link scheduling problems mentioned in the introduction to further illustrate the items in the formulations in (3) and (4). One of the examples also gives more details for the experiment in Section 4.2.
>
> > But then, in the MDP case, we have also f_obj and r. I understand now that we are going to describe a policy by specifying a cost function c and a heuristic function. I also understand, I guess, that we are trying, ultimately, to optimize f_obj (in 4a). But: o I don't understand how that relates to f_r o I'm not sure I see why it makes sense to apply f_obj to the expected (averaged over time?) o values?
>
> To address these concerns, we updated the objective function in the formulation of R-COP in graph-based MDP in (4a) and (10a), and subsequently (11) and the discussions before and after (11), as well as lines 4, 21 in Algorithm 2 in the Appendix D (former Appendix C).
>
> The updated objective function is essentially the expected value of the initial state over its distribution, capturing two types of objective functions: the average reward and start-state formulations, mentioned in key literatures, such as policy gradient in (Sutton et al., 1999) and deterministic policy gradient algorithm in (Silver et al., 2014). The updated objective function takes the form of start-state formulation (expected cumulative discounted rewards), but can also be instantiated as average reward, as illustrated by the second exemplary formulation in Appendix A (delay-oriented link scheduling).
>
> The update does not require any changes to the deductions or core procedures. However, it captures the true objective of MDP more accurately, and is consistent with previous work (Silver et al., 2014, Sutton et al., 1999). In this way, it makes the paper much easier to follow, and the readers can relate the rewards and $f_r$ to the objective function more easily. The updated objective can also be better related to the procedure of Algorithm 2 in Appendix D.
>
> > Section 3.1 is hard to decode. In the end, is the idea that the Z values are parameters of h (indirectly, via f_loc)? And then this is a straight-up optimization problem (not sequential)?
>
> We added two sentences to the discussion after (7) in Section 3.1 to explain that: for applications on static topologies, we can directly optimize Z without using an actor network (in this case it is a straight-up optimization problem); However, for applications on dynamic topologies, we want the Z values being a function of the topology (implemented by an actor network), allowing quicker adaptation to topological changes.
>
> > In the experimental section, begin by listing the claims you want to make about your method, and then provide clear and specific data and arguments supporting those claims. In the discussion of the number of rounds for convergence---is this about training time or performance time? It's not at all clear what these "rounds" are.
>
> We added the purposes of experiments at the beginning of Section 4.1 and 4.2. In addition, we updated the title of Figure 3 to explain the "rounds" and that the figure is for performance time. We also made it clear that the discussion in Section 4.3 is about training time.